# $N^6$-methyladenosine regulates glycolysis of cancer cells through PDK4

Zihan Li[1,2,6], Yanxi Peng[1,3,6], Jiexin Li[1,6], Zhuojia Chen[4], Feng Chen[1], Jian Tu[2], Shuibin Lin ●[5] & Hongsheng Wang ●[1✉]

Studies on biological functions of $N^6$-methyladenosine (m$^6$A) modification in mRNA have sprung up in recent years. We find m$^6$A can positively regulate the glycolysis of cancer cells. Specifically, m$^6$A-sequencing and functional studies confirm that pyruvate dehydrogenase kinase 4 (PDK4) is involved in m$^6$A regulated glycolysis and ATP generation. The m$^6$A modified 5′UTR of *PDK4* positively regulates its translation elongation and mRNA stability via binding with YTHDF1/eEF-2 complex and IGF2BP3, respectively. Targeted specific demethylation of *PDK4* m$^6$A by dm$^6$ACRISPR system can significantly decrease the expression of PDK4 and glycolysis of cancer cells. Further, TATA-binding protein (TBP) can transcriptionally increase the expression of Mettl3 in cervical cancer cells via binding to its promoter. In vivo and clinical data confirm the positive roles of m$^6$A/PDK4 in tumor growth and progression of cervical and liver cancer. Our study reveals that m$^6$A regulates glycolysis of cancer cells through PDK4.

[1] Guangdong Key Laboratory of Chiral Molecule and Drug Discovery, School of Pharmaceutical Sciences, Sun Yat-sen University, Guangzhou, Guangdong 510006, China. [2] Institute of Pharmacy & Pharmacology, University of South China, Hengyang, Hunan 421001, China. [3] Department of Basic Medicine, Xiangnan University, Chenzhou, Hunan 423000, China. [4] Sun Yat-sen University Cancer Center; State Key Laboratory of Oncology in South China; Collaborative Innovation Center for Cancer Medicine, Guangzhou 510060, China. [5] Center for Translational Medicine, The First Affiliated Hospital, Sun Yat-sen University, Guangzhou, China. [6] These authors contributed equally: Zihan Li, Yanxi Peng, Jiexin Li. ✉ email: whongsh@mail.sysu.edu.cn

N⁶-methyladenosine (m⁶A) is the prominent dynamic mRNA modification which has been identified since the 1970s[1]. m⁶A is governed by methyltransferase complex ("writers"), demethylases ("erasers"), and RNA-binding proteins ("readers")[2]. Methyltransferase-like 3 (Mettl3), Methyltransferase-like 14 (Mettl14), and Wilms' tumor 1-associating protein (WTAP) form the core methyltransferase complex[3,4]. Two demethylases are fat mass and obesity-associated protein (FTO) and AlkB homologue 5 (ALKBH5)[5,6]. m⁶A can regulate the splicing, translation, and decay rates of mRNA to affect protein production and various biological processes such as cell differentiation, embryonic development, and stress responses[7].

Dysregulation of m⁶A methylated mRNA can regulate cancer development. FTO has been suggested to play oncogenic role in acute myeloid leukemia[8,9], while inhibition of FTO exerts a broad anti-leukemic activity in vitro and in vivo[10]. ALKBH5 can maintain tumorigenicity of glioblastoma stem-like cells via sustaining FOXM1 expression[11]. Our recent study showed that deletion of Mettl3 can suppress the epithelial to mesenchymal transition (EMT) of cancer cells by inhibiting translation of Snail[12]. Mettl3 can directly interact with eukaryotic translation initiation factor 3 subunit h (eIF3h) to promote tumorigenicity[13]. Although relationship between m⁶A modification and tumorigenesis has been raised in recent years, the precise mechanisms have not been sufficiently described.

Recently, cancer metabolism is gaining increasing attention in cancer research. As a hallmark of cancer cells, Warburg effect means cancer cells heavily rely on glycolysis to obtain energy regardless of the presence of oxygen[14]. There is a strong link between epigenetic factors and the Warburg effect. For example, cancer cells can adopt the metabolic diversion through chromatin remodeling and histone acetylation (H3K27ac)[15]. Studies uncover that there might be clues between RNA modification and cellular metabolism. Transcriptome-wide m⁶A profiling revealed that genes relevant to regulation of metabolic processes were modified by m⁶A during liver development[16]. Knockdown of Mettl3 can reduce lipid accumulation and cell viability[17]. However, the roles of m⁶A modification in regulating mRNA processing of metabolic-related genes and affecting metabolism of cancer cells remain unknown.

Our present study reveals that m⁶A can positively regulate the glycolysis of cancer cells via regulation of pyruvate dehydrogenase kinase 4 (PDK4), one of the most important factors which can direct carbon flux into glycolysis from oxidative phosphorylation (OXPHOS). The methylation of 5′UTR of PDK4 can regulate the mRNA stability and translation of PDK4 via recruitment of different reader proteins.

## Results

**m⁶A regulates glycolysis and ATP generation of cancer cells.** To investigate the role of m⁶A modification in cell metabolism, we used Mettl3^{Mut/-} HeLa cells generated in the previous study[18] by using CRISPR/Cas9 editing system and Mettl3 stable knockdown (KD) Huh7, HepG2, and MDA-MB-231 cells[12] by using sh-RNA (Supplementary Fig. 1A and B). Mettl3^{Mut/-} HeLa and sh-Mettl3 Huh7 cells exhibited significantly lower glucose consumption (Fig. 1a), lactate production rate (Fig. 1b), and the ATP levels (Fig. 1c) than that of control cells. Knockdown of Mettl3 can inhibit the ATP generation of both HepG2 and MDA-MB-231 cells (Supplementary Fig. 1C). Further, Mettl3^{Mut/-} HeLa cells displayed decreased extracellular acidification rate (ECAR), which reflects overall glycolytic flux, and increased oxygen consumption rate (OCR), an indicator of mitochondrial oxidative respiration (Fig. 1d, e). Consistently, increased OCR while decreased ECAR were observed in sh-Mettl3 Huh7 cells as compared with that in control cells (Fig. 1f, g).

To confirm the role of m⁶A in cell metabolism, we generated catalytically inactive Mettl3 mutant DA (D395A)[19] and observed that cells expressing wild-type (WT) Mettl3, rather than Mettl3 DA mutant (Supplementary Fig. 1D), displayed an increased glucose consumption (Supplementary Fig. 1E), lactate production (Supplementary Fig. 1F), and ATP productions (Supplementary Fig. 1G). Over expression of m⁶A demethylase ALKBH5 can also decrease the glucose consumption, lactate production rate, and ATP levels of Huh7 cells (Fig. 1h). Consistently, ALKBH5 can increase OCR while decrease ECAR in HeLa cells (Supplementary Fig. 1H and I). Together, our findings reveal that m⁶A positively regulated glycolysis and ATP generation of cervical and liver cancer cells.

**PDK4 mediates m⁶A regulated glycolysis and ATP generation.** We checked whether m⁶A regulated metabolic reprogramming is related to the mitochondrial mass of cancer cell. The results suggested that deletion of Mettl3 had no significant effect on mitochondrial DNA content (Supplementary Fig. 2A). As Mettl3 is an m⁶A "writer" of RNA methylation, we next investigated potential targets involved in m⁶A-regulated metabolic shift by use of m⁶A sequencing (m⁶A-seq)[12] and mRNA-seq. Gene ontology (GO) analysis of these downregulated genes showed enrichment in several key ontology terms reflecting cellular metabolic processes (Fig. 2a). Gene set enrichment analysis (GSEA) found that the glycolysis (Fig. 2b) and glycan degradation (Supplementary Fig. 2B) were inhibited in Mettl3^{Mut/-} HeLa cells. Among the 83 glucose metabolism related genes (Supplementary Table 1), we identified the only candidate, pyruvate dehydrogenase kinase 4 (PDK4, Supplementary Table 2), that overlapping among mRNA-seq (greater than 2.0-fold variation ($p < 0.05$) between wild type and Mettl3^{Mut/-} HeLa cells, Supplementary Data 1) and m⁶A-seq (modification is more than 3 times greater than that in the input, $p < 0.05$, Supplementary Data 2) (Fig. 2c).

PDK4 has been suggested as one of the most important factors controlling cell metabolism via directing carbon flux into glycolysis from OXPHOS[20]. Our data confirmed that PDK4 mRNA can be m⁶A methylated and showed significant enrichment of m⁶A in its 5′UTR and 3′UTR regions (Fig. 2d), which is consistent with published reports using human HEK293T[21], A549[22], and GM12878[23] cells. m⁶A-RIP-qPCR confirmed that a 6-fold m⁶A antibody enriched PDK4 mRNA in HeLa cells, while this enrichment significantly decreased in Mettl3^{Mut/-} HeLa cells (Fig. 2e). Consistently, knockdown of Mettl3 significantly attenuated m⁶A antibody enriched PDK4 mRNA in Huh7 cells (Fig. 2f). However, neither m⁶A pull down enrichment nor m⁶A downregulation in Mettl3^{Mut/-} HeLa cells was observed for the negative candidate gene PDK1 (Supplementary Fig. 2C).

Our data showed that expression of PDK4 was decreased in Mettl3 knocked down HeLa and Huh7 cells (Fig. 2g). Consistently, knockdown of Mettl3 can decrease the expression of PDK4 in SiHa cells (Supplementary Fig. 2D). While over expression of Mettl3 increased the protein expression of PDK4 in both HeLa and MDA-MB-231 cells (Fig. 2h). Further, qRT-PCR confirmed that the mRNA of PDK4 was also decreased in Mettl3 knockdown cells (Fig. 2i). However, the mRNA expression of PDK1/2/3 was not variated in Mettl3^{Mut/-} cells (Supplementary Fig. 2E). Over expression of ALKBH5 can suppress the expression of PDK4 in both HeLa and Huh7 cells (Fig. 2g). These data indicated that m⁶A can positively regulate the expression of PDK4 in cancer cells.

Although the promotion roles of PDK4 in cell metabolism have been well illustrated[24,25], we further investigated its roles in m⁶A regulated glycolysis of cancer cells. Results showed that over expression of PDK4 (Supplementary Fig. 2F) to a comparable level

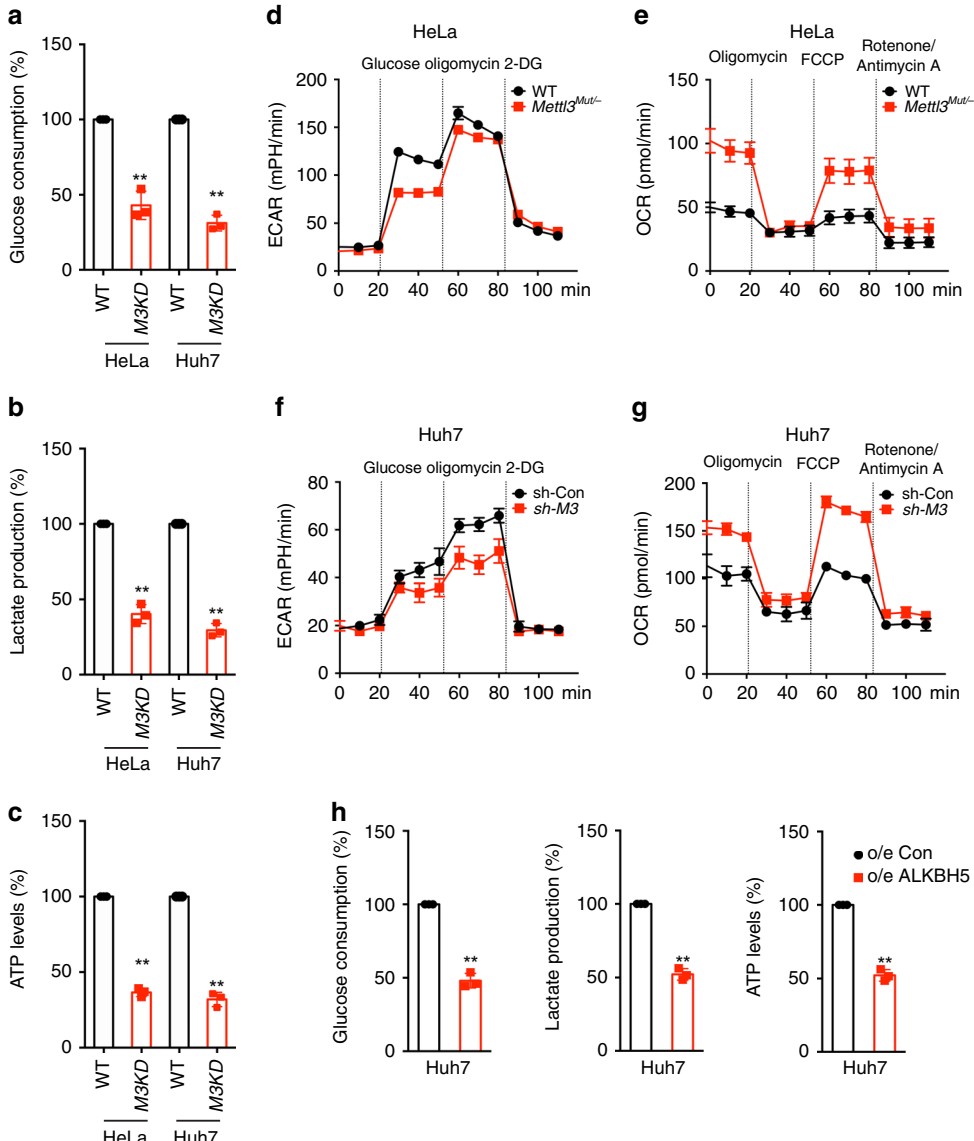

**Fig. 1 m⁶A regulates glycolysis and ATP generation of cancer cells. a–c** The glucose consumption (**a**), lactate production (**b**), and ATP levels (**c**) in *Mettl3^Mut/-* HeLa, sh-*Mettl3* Huh7 and their corresponding control cells. **d**, **e** The cellular ECAR (**d**) and OCR (**e**) were measured in wild type and *Mettl3^Mut/-* HeLa cells. **f**, **g** The cellular ECAR (**f**) and OCR (**g**) were measured in sh-control and sh-*Mettl3* Huh7 cells. **b** The glucose consumption, lactate production, and ATP levels in Huh7 cells transfected with vector control or ALKBH5 constructs. Data are presented as the mean ± SD from three independent experiments. \*\*$p < 0.01$ by two tailed t test for (**a**) ($p = 0.005$ and $p < 0.0001$), (**b**) ($p < 0.0001$), (**c**) ($p < 0.0001$), and (**h**) ($p < 0.0001$).

of endogenous PDK4 can predominantly reverse the glucose consumption, lactate production rate, and ATP levels in *Mettl3^Mut/-* cells (Fig. 2j). Consistently, over expression of PDK4 constructs to a comparable level of endogenous PDK4 (Supplementary Fig. 2F) can also attenuate the metabolic phenotypes in Mettl3 knocked down SiHa cells (Supplementary Fig. 2G). It confirmed that PDK4 was involved in Mettl3 regulated glycolysis of cervical and liver cancer cells.

**m⁶A regulates mRNA stability and translation of PDK4.** We further investigated the potential mechanisms involved in m⁶A regulated expression of PDK4. Our data showed that there was no significant difference of promoter activities of *PDK4* between wild type and *Mettl3^Mut/-* HeLa cells, or between sh-control and sh-*Mettl3* Huh7 cells (Fig. 3a), suggesting that m⁶A did not affect the transcription of *PDK4*. After fractionation assay (Supplementary Fig. 3A), results showed that there was no difference of subcellular localization of *PDK4 mRNA* between wild type and

*Mettl3^Mut/-* cells, or between sh-control and sh-*Mettl3* Huh7 cells (Fig. 3b). We treated wild type and *Mettl3^Mut/-* HeLa cells with Act-D to block transcription. Deletion of Mettl3 had no significant effect on the stability of precursor mRNA of *PDK4* in HeLa cells, suggesting that m⁶A had no effect on splicing of *PDK4* precursor mRNA (Fig. 3c). However, mRNA stability of mature *PDK4* in Mettl3 deletion HeLa (Fig. 3d) or SiHa (Supplementary Fig. 3B) cells were significantly less than that in their corresponding control cells. Consistently, over expression of ALKBH5 can also decrease the mature mRNA stability of *PDK4* in HeLa cells (Supplementary Fig. 3C). It indicated that m⁶A modification may delay the degradation of mature mRNA of *PDK4* in cancer cells.

We further investigated whether m⁶A can regulate the expression of PDK4 beside mRNA stability. Both wild-type HeLa cells transfected with vector or Mettl3 construct were further treated with MG132 to inhibit proteasome activity or cycloheximide (CHX) to block protein translation. The data showed that

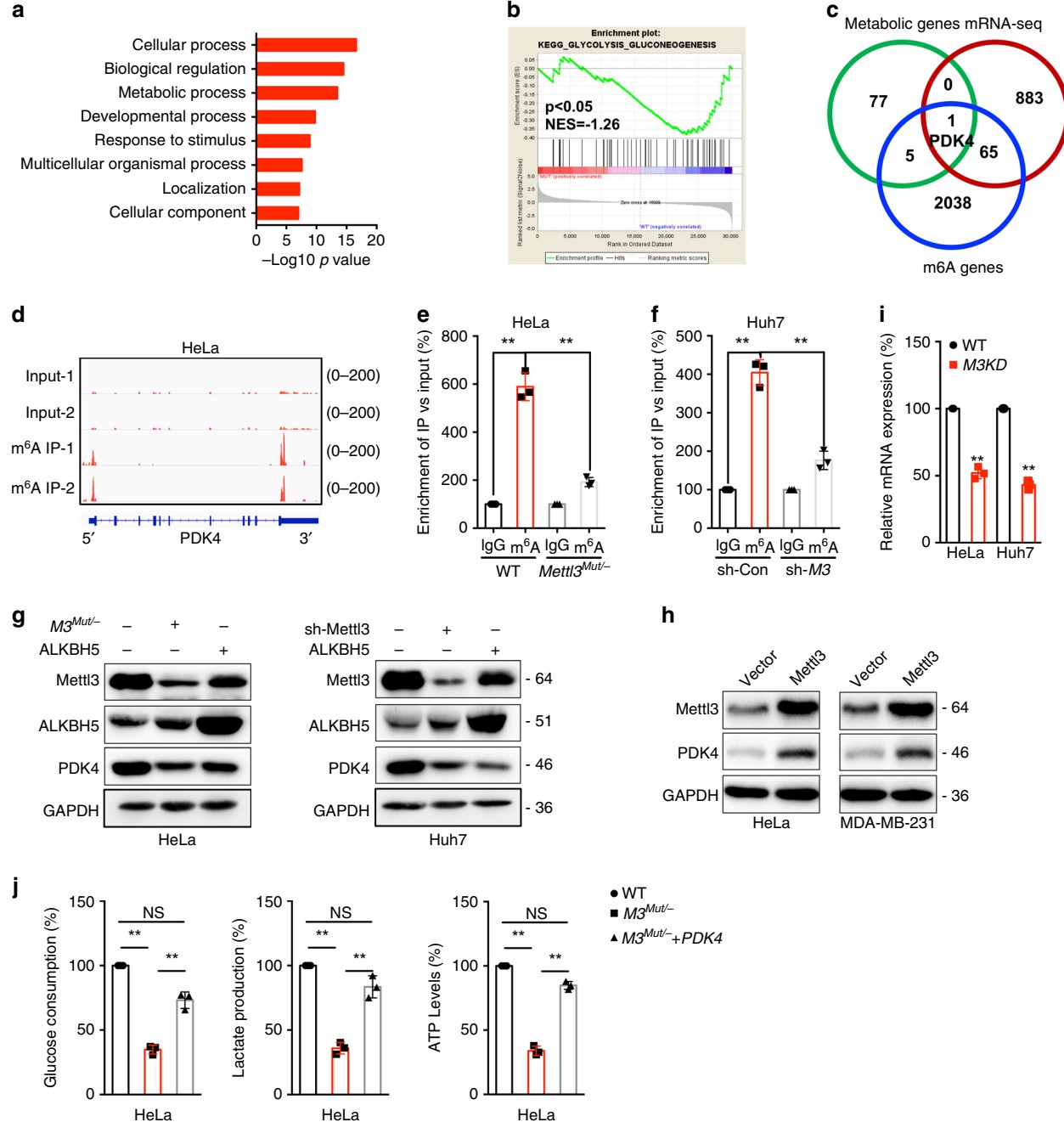

**Fig. 2 PDK4 mediates m⁶A regulated glycolysis and ATP generation of cancer cells. a** Gene ontology analysis was performed on a subset of downregulated genes in *Mettl3^Mut/-* HeLa cells. Log2 fold change, (KD: WT) < −0.5, was applied as the threshold cutoff. Several biological processes involved in metabolic processes were enriched and highlighted in bold; Significance shown as −Log10 Bonferroni *p*-value after multiple hypothesis correction. **b** GSEA reveals negative enrichment of genes in glycolysis gluconeogenesis sets of *Mettl3^Mut/-* HeLa cells. **c** Venn diagram shows substantial and significant overlap among metabolic genes, variated genes in *Mettl3^Mut/-* HeLa cells (>2 folds), and m⁶A enriched genes in wild type HeLa cells (>3 folds than input). **d** m⁶A peaks were enriched in 5′UTR and 3′UTRs of *PDK4* genes from m⁶A RIP-seq data; **e** m⁶A RIP-qPCR analysis of *PDK4 mRNA* in wild type and *Mettl3^Mut/-* HeLa cells. **f** m⁶A RIP-qPCR analysis of *PDK4 mRNA* in sh-Con and sh-*Mettl3* Huh7 cells. **g** The expression of PDK4 in *Mettl3^Mut/-* HeLa, sh-Mettl3 Huh7, or over expression of ALKBH5 and their corresponding control cells were measured by western blot analysis. **h** Cells were transfected with vector control or Mettl3 construct for 24 h, the expression of PDK4 was measured. **i** The mRNA of PDK4 in *Mettl3^Mut/-* HeLa, sh-*Mettl3* Huh7 and their corresponding control cells were measured by qRT-PCR. **j** The glucose consumption, lactate production, and ATP levels in wild type or *Mettl3^Mut/-* HeLa cells transfected with PDK4 constructs for 24 h. Data are presented as the mean ± SD from three independent experiments. A representative from a total of two to three independent experiments is shown for (**g**) and (**h**). **\*\****p* < 0.01, NS, no significant, by random permutation test for (**b**), by two-way ANOVA for (**e**) (*p* = 0.0001 and *p* = 0.0004, respectively) and (**f**) (*p* < 0.0001 and *p* = 0.0007, respectively), two-tailed unpaired Student's *t* test for (**i**) (*p* < 0.0001), and one-way ANOVA for (**j**) (*p* < 0.0001 and *p* = 0.0008 for glucose consumption, *p* < 0.0001 and *p* = 0.0011 for lactate production, and *p* < 0.0001 for ATP levels, respectively).

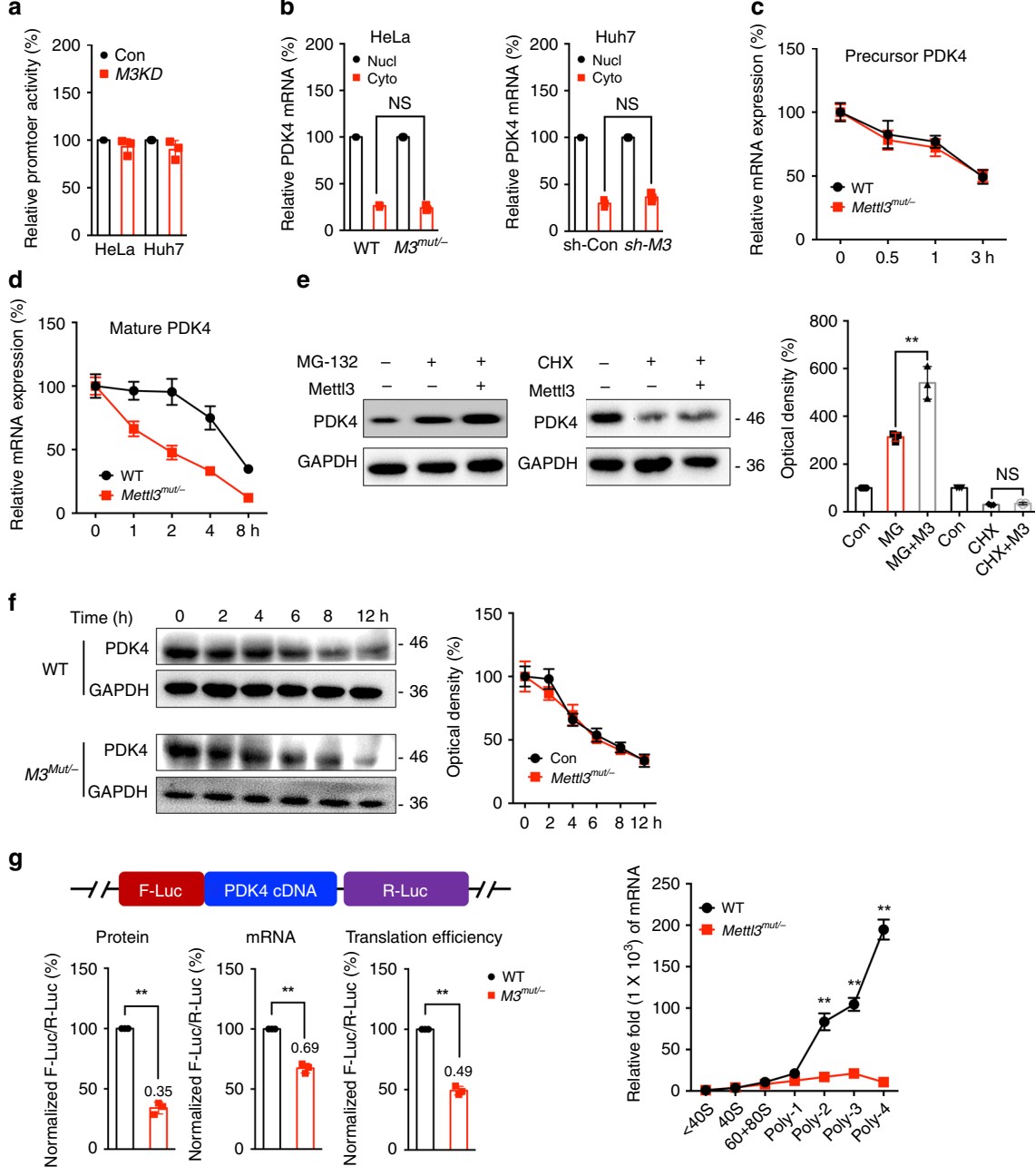

**Fig. 3 m⁶A regulates the mRNA stability and translation of PDK4 in cancer cells. a** Cells were transfected with pGL3-Basic-PDK4-luc reporter and pRL-TK plasmid for 24 h. Results were presented as the ratios between the activity of the reporter plasmid and pRL-TK. **b** The relative levels of nuclear versus cytoplasmic *PDK4* mRNA in wild-type and *Mettl3*$^{Mut/-}$ cells, or sh-control and sh-*Mettl3* Huh7 cells. **c** After treatment with Act-D to inhibit transcription, the precursor mRNA levels of *PDK4* were checked in wild type and *Mettl3*$^{Mut/-}$ cells. **d** After treatment with Act-D for the indicated times, the mature mRNA levels of *PDK4* were checked in wild type and *Mettl3*$^{Mut/-}$ cells. **e** HeLa cells were pre-transfected with vector control or Mettl3 construct for 24 h and then further treated with CHX (10 μg/ml) or MG-132 (5 μM) for 6 h, the expression of PDK4 was detected by western blot analysis (left) and quantitatively analyzed (right). **f** Cells were treated with 10 μg/ml CHX for the indicated time periods, the expression of PDK4 was detected by western blot analysis (left) and quantitatively analyzed (right). **g** Wild-type or *Mettl3*$^{Mut/-}$ HeLa cells were transfected with pmirGLO-PDK4 reporter for 24 h. The translation outcome was determined as a relative signal of F-luc divided by R-luc, the mRNA abundance was determined by qRT-PCR of F-luc and R-luc, and the translation efficiency of PDK4 is defined as the quotient of reporter protein production (F-luc/R-luc) divided by mRNA abundance[28]. **h** qRT-PCR checked the mRNA levels of PDK4 in non-ribosome portion (<40S), 40S, 60S, 80S, and polysome fractions in wide type and *Mettl3*$^{Mut/-}$ HeLa cells. Data are presented as the mean ± SD from three independent experiments. *$p < 0.05$, **$p < 0.01$, NS, no significant, by two-tailed unpaired Student's *t* test for (**a**) and (**c**) ($p < 0.0001$ for all comparisons), and two-way ANOVA for (**b**) ($p > 0.05$), and by one-way ANOVA for (**e**) ($p = 0.005$) and (**h**) ($p < 0.0001$ for all comparisons).

in the presence of CHX, but not MG-132, attenuated Metttl3-induced PDK4 expression in HeLa cells (Fig. 3e), suggesting that Mettl3 might regulate the protein translation rather than protein stability/post-translation modification of PDK4. This was confirmed by the results that half-life of PDK4 protein had no significant difference between wide type and $Mettl3^{Mut/-}$ HeLa cells (Fig. 3f).

To confirm that m6A may regulate the translation of PDK4, we constructed the pmirGLO-PDK4 luciferase reporter that contained PDK4 cDNA in multiple cloning site (MCS) region. Results of dual-luciferase assay showed that translation efficiency of PDK4 in HeLa cells was significantly greater than in $Mettl3^{Mut/-}$ HeLa cells (Fig. 3g). Consistently, the translation efficiency of PDK4 was also reduced in cells co-transfected with ALKBH5 constructs (Supplementary Fig. 3D). It suggested that m6A-induced PDK4 expression might be due to regulation of translation. Next, we isolated different RNA fractions from wild type and $Mettl3^{Mut/-}$ HeLa cells by ribosome profiling: non-translating fraction (<40S), translation initiation fraction (including 40S ribosomes, 60S ribosomes, 80S monosomes, <80S) and translation active polysomes (>80S) (Supplementary Fig. 3E). Results showed that PDK4 mRNA in translation active polysomes (>80S), rather than that in monosome (<40S, 40S, 60S, and 80S), of $Mettl3^{Mut/-}$ cells were significantly lower than that in wild type cells (Fig. 3h). All these data suggested that m6A regulated the mRNA stability and translation elongation of PDK4 in cervical and liver cancer cells.

**Methylation sites involved in m6A-regulated PDK4**. m6A-RIP-seq data showed that one m6A peak in 5′UTR region and three peaks in 3′UTR region of PDK4 mRNA (Fig. 2d and Fig. 4a). To characterize m6A methylation in PDK4 mRNA, fragmented RNA isolated from HeLa cells was immunoprecipitated with m6A antibody[12]. m6A-RIP-PCR showed that the m6A enrichment of PDK4 5′UTR, rather than 3′UTR, was significantly downregulated in $Mettl3^{Mut/-}$ cells (Fig. 4b), suggesting that m6A methylation in 5′UTR region might be more dynamic than those in 3′UTR.

To explore the potential roles of m6A methylation on PDK4 expression, firstly, we constructed 3′UTR reporters containing wild type PDK4 3′UTR, or mutant 1/2/3 3′UTR (GGAC to GGCC, Fig. 4c) after the firefly luciferase reporter gene. The luciferase assay showed that the mRNA expression, protein expression, and translation efficiency of pmirGLO-PDK4-3′UTR in $Mettl3^{Mut/-}$ HeLa cells had no significant difference with that in HeLa cells (Fig. 4d). The PDK4 3′UTR-reporter luciferase assay showed that reporter genes from WT-3′UTR, mut1-3′UTR, mut2-3′UTR, and mut3-3′UTR, had no translational difference between wild type and $Mettl3^{Mut/-}$ cells (Fig. 4e). It suggested that m6A methylation on 3′UTR was independent to the m6A -regulated PDK4 expression.

To investigate the potential roles of m6A in 5′UTR, we generated luciferase reporters by inserting 5′UTR of PDK4 in the front of firefly luciferase (Fig. 4f). The dual-luciferase assay showed that the mRNA expression, protein expression, and translation efficiency of pGL3-PDK4-5′UTR in $Mettl3^{Mut/-}$ HeLa cells were significantly less than that in wild type HeLa cells (Fig. 4g). The mutant of m6A motif (5′UTR-Mut1) while not control site (5′UTR-Mut2) in 5′UTR resulted in a downregulation of mRNA expression, protein expression, and translation efficiency of F-Luc but partially abolished the difference between wild type and $Mettl3^{Mut/-}$ cells (Fig. 4g). Wild type or $Mettl3^{Mut/-}$ HeLa cells were transfected with pcDNA-PDK4-5′UTR-WT or pcDNA-PDK4-5′UTR-Mut1/Mut2, respectively. Western blot results showed that deletion of Mettl3-suppressed PDK4 expression was attenuated using pcDNA-PDK4-5′UTR-Mut1, compared to that of pcDNA-PDK4-5′UTR-WT or pcDNA-PDK4-5′

UTR-Mut2 (Fig. 4h). This was further confirmed by the results that the mRNA stability of pcDNA-PDK4-5′UTR-WT in wild type cells were greater than that in $Mettl3^{Mut/-}$ cells (Fig. 4i), while pcDNA-PDK4-5′UTR-Mut1 can abolish the difference of mRNA half-lives between wild type and $Mettl3^{Mut/-}$ cells (Fig. 4j). Together, our data suggested that m6A in PDK4 5′UTR is responsible for the mRNA stability and translation, which might be due to that the m6A modified GGAC is important for the secondary structure of PDK4 5′UTR (Supplementary Fig. 4).

**Factors involved in m6A regulated expression of PDK4**. Mechanisms responsible for m6A regulated mRNA stability and translation elongation were further investigated. It has been revealed that m6A modification may regulate the mRNA stability via readers including YTHDF2, YTHDF3, and IGF2BP1~3[26,27]. Results showed that IGF2BP3, while not YTHDF2, YTHDF3, or IGF2BP1/2, can significant bind with PDK4 mRNA in HeLa cells (Fig. 5a). Further, the binding between IGF2BP3 and PDK4 was significantly decreased in $Mettl3^{Mut/-}$ cells as compared with that in HeLa cells (Fig. 5b). We therefore knocked down the expression of IGF2BP3 by siRNA (Fig. 5c). Results showed that si-IGF2BP3 can suppress the expression of PDK4 and attenuate deletion of Mettl3-suppressed expression of PDK4 in HeLa cells (Fig. 5c). Further, si-IGF2BP3 can significantly decrease the mRNA stability of pcDNA-PDK4-5′UTR-WT (Fig. 5d), while this effect was attenuated for pcDNA-PDK4-5′UTR-Mut1 (Fig. 5i). The data suggested that IGF2BP3 was involved in m6A regulated mRNA stability.

YTHDF1 can recognize m6A methylated mRNA and promote the translation of its targets[28]. RIP-qPCR was used to verify whether YTHDF1 participates in m6A methylation of PDK4 mRNA. Results showed that YTHDF1 interacted with PDK4 mRNA remarkably, while this interaction was significantly suppressed in $Mettl3^{Mut/-}$ cells (Fig. 5f). It further revealed that YTHDF1 preferred to interact with the 5′UTR region of PDK4 mRNA rather than 3′UTR region (Fig. 5g). In $Mettl3^{Mut/-}$ cells, interaction between YTHDF1 and 5′UTR, but not 3′UTR, of PDK4 mRNA, was significantly suppressed in comparison to that in HeLa cells (Fig. 5g). To confirm the roles of YTHDF1 in m6A-regulated PDK4 expression, we over expressed YTHDF1 in HeLa cells (Fig. 5h). Our data showed that YTHDF1, which had limited effect on the mRNA expression of PDK4 (Supplementary Fig. 5A), can increase the expression of PDK4 and attenuate deletion of Mettl3-suppressed expression of PDK4 in HeLa cells (Fig. 5h). Further, over expression of YTHDF1 can increase the protein expression of pcDNA-PDK4-5′UTR-WT, while this effect was attenuated for pcDNA-PDK4-5′UTR-Mut1 (Fig. 5i).

To further investigate whether m6A regulated translation of PDK4 was cap-independent/dependent translation initiation, both $Mettl3^{Mut/-}$ and wide type cells were treated with rapamycin, which inhibits cap-dependent translation[29]. However, no obvious difference of PDK4 was observed between wild type and $Mettl3^{Mut/-}$ cells (Supplementary Fig. 5B). Maintenance of PDK4 expression in the presence of rapamycin was not due to an increase of PDK4 protein stability, as co-treatment of wild type cells with protein synthesis inhibitor CHX resulted in a rapid disappearance of PDK4 (Supplementary Fig. 5B), suggesting that PDK4 expression was regulated through cap-independent translation[30].

We then further evaluated the roles of translation elongation in m6A regulated PDK4 according the results of ribosome profiling-qPCR. eEF-1 and eEF-2 are elongation factors that responsible to the eukaryotic translation elongation[31]. RIP-qPCR was performed to investigate the variation of eEF-1 and eEF-2 binding PDK4 mRNA. Our data revealed that the interaction between PDK4 mRNA and eEF-2, but not eEF-1, in $Mettl3^{Mut/-}$ HeLa cells was

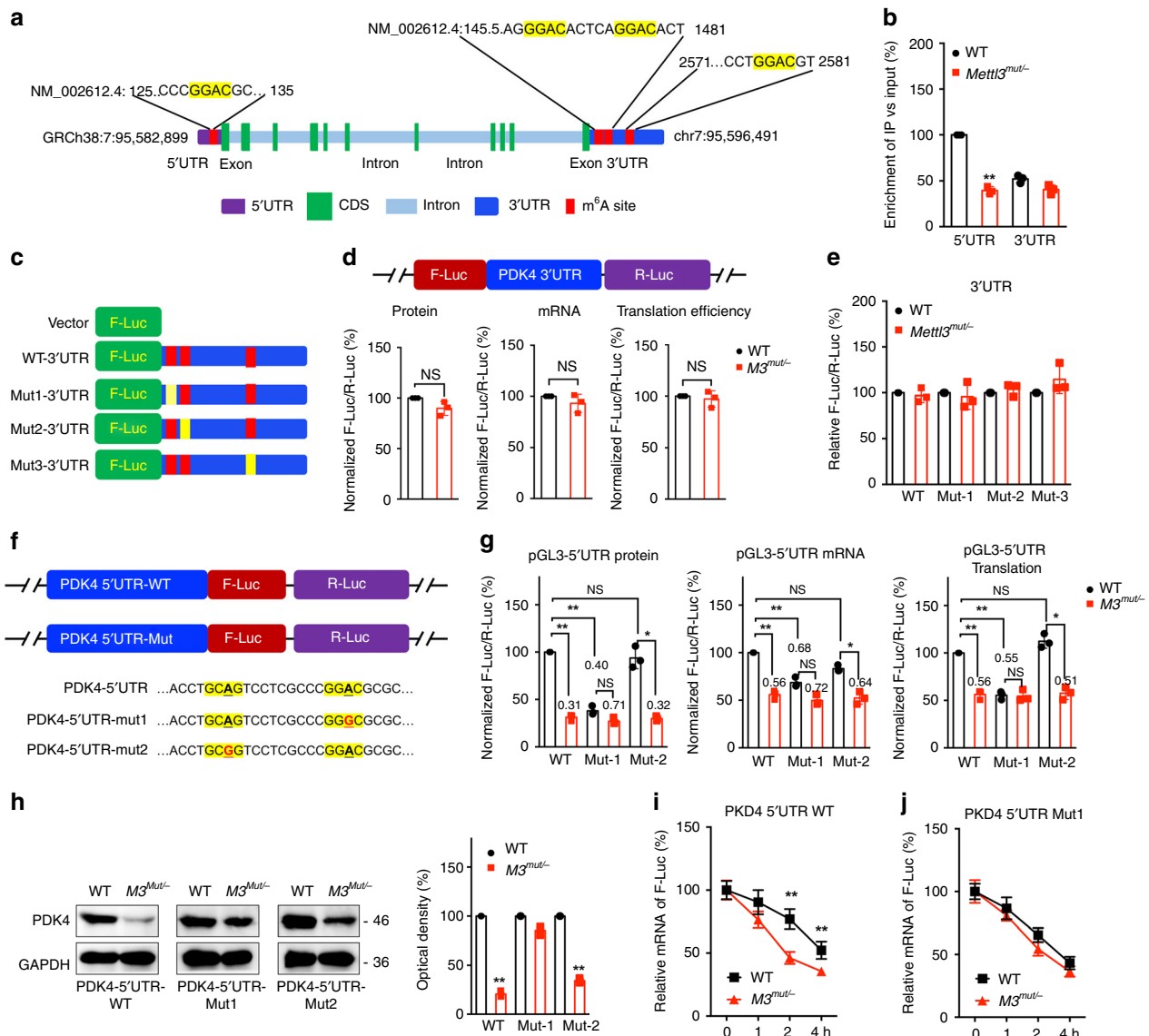

**Fig. 4 Methylation sites of PDK4 involved in m⁶A regulated expression of PDK4. a** Schematic representation of positions of m6A motifs within *PDK4* mRNA. **b** The m6A in 5′UTR or 3′UTR of PDK4 in wild type or *Mettl3^Mut/-* HeLa cells were analyzed by m6A-RIP-qPCR using fragmented RNA. **c** Schematic representation of mutated (GGAC to GGCC) 3′UTR of pmirGLO vector to investigate the roles of m6A in 3′UTR in PDK4 expression. **d** Wild-type or *Mettl3^Mut/-* HeLa cells were transfected with pmirGLO-PDK4-3′UTR-WT reporter for 24 h. The protein, mRNA and translation efficiency were determined. **e** The relative luciferase activity of F-Luc/R-Luc of pmirGLO-3′UTR-WT, or pmirGLO-3′UTR-Mut-1/-2/-3 in wild type and *Mettl3^Mut/-* HeLa cells; **f** Schematic representation of mutation in 5′UTR to investigate the m6A roles on PDK4 expression. **g** Wild-type or *Mettl3^Mut/-* HeLa cells were transfected with pGL3-PDK4-5′UTR-WT or pGL3-PDK4-5′UTR-Mut1/2 reporter for 24 h. The protein, mRNA and translation efficiency were determined. **h** pcDNA-PDK4-5′UTR-WT or pcDNA-PDK4-5′UTR-Mut1/2 was transfected into wild type or *Mettl3^Mut/-* HeLa cells for 24 h. Protein expression was measured by western blot analysis (left) and quantitatively analyzed (right). **i, j** pcDNA-PDK4-5′UTR-WT (**i**) or pcDNA-PDK4-5′UTR-Mut1 (**j**) was transfected into wild type or *Mettl3^Mut/-* HeLa cells for 24 h and then further treated with Act-D for the indicated times. The mRNA of PDK4 was checked by qRT-PCR. Data are presented as the mean ± SD from three independent experiments. *$p < 0.05$, **$p < 0.01$, NS, no significant, by two-tailed unpaired Student's $t$ test for (**b**) ($p < 0.0001$), (**d**) ($p > 0.05$), and (**h**) ($p < 0.0001$ for all), and two-way ANOVA for (**c**) and (**i**) ($p < 0.001$ for all).

significantly inhibited than that in HeLa cells (Fig. 5j). Combining our previous data that the YTHDF1 and eEF-2 interaction, but not eEF-1, significantly suppressed in *Mettl3^Mut/-* HeLa cells than that in control cells[12], the data hinted that both YTHDF1 and eEF-2 are likely associated to regulate the m6A induced translation elongation of *PDK4 mRNA* in cancer cells.

**Targeting m⁶A of PDK4 by dm⁶ACRISPR to regulate glycolysis**. We then specifically demethylated the m6A of PDK4 by fusing the catalytically dead Type VI-B Cas13 enzyme with the

m6A demethylase ALKBH5, which is developed in our lab and named as dm6ACRISPR[32]. Three gRNAs at distinct positions around the m6A site were designed to target the mRNA of PDK4 (Fig. 6a). Firstly, the m6A site in 5′UTR of PDK4 was verified by SELECT method[33] in HeLa and Huh7 cells (Fig. 6b). Next, wild type Cas13b, which cleaves targeted mRNA, was used to test the efficiency of gRNAs. Our data showed that wild type Cas13b co-transfected with three gRNAs respectively can significantly decrease the mRNA levels of *PDK4* (Supplementary Fig. 6A), suggesting that all three gRNAs worked efficiently in vivo. While mRNA levels of PDK4 in cells transfected with gRNAs alone

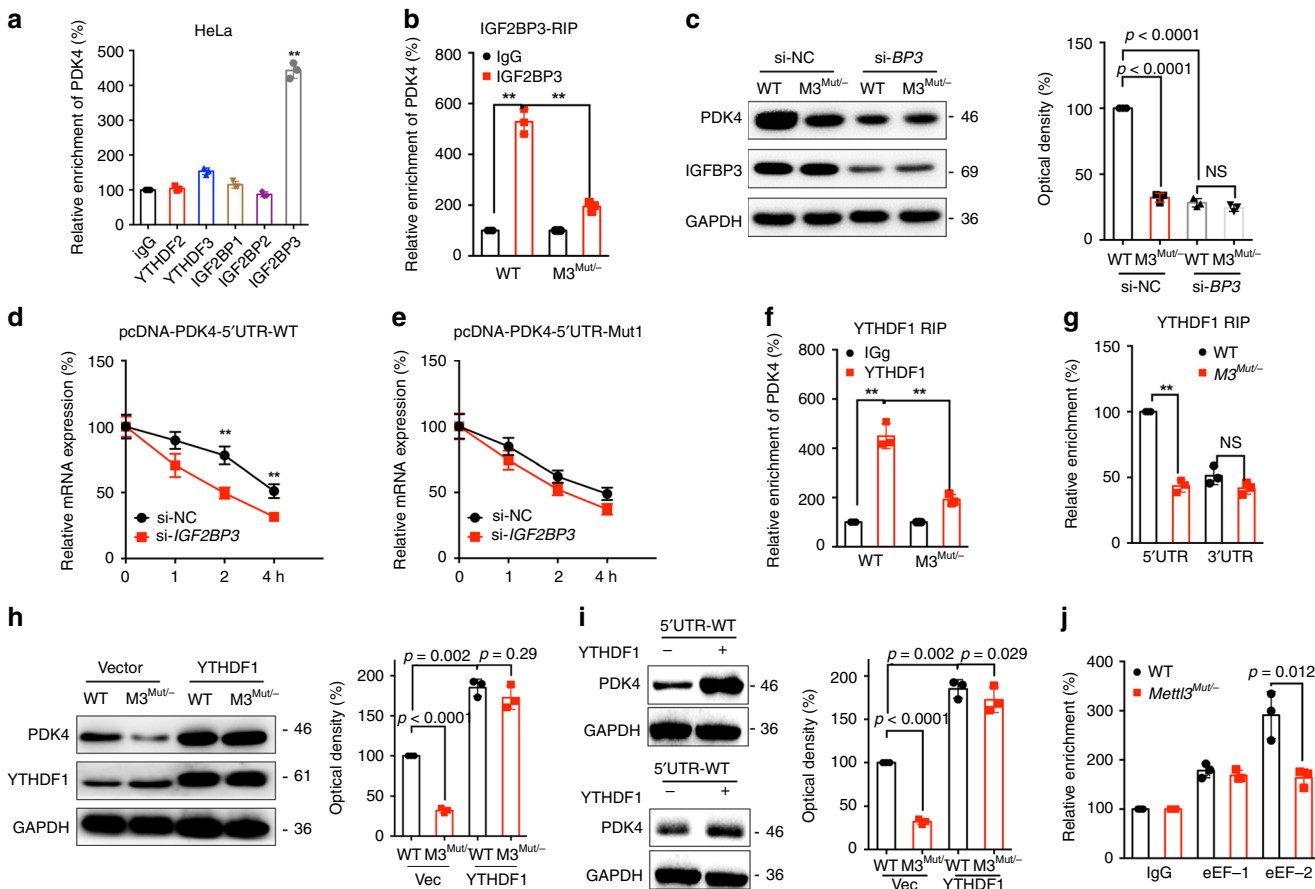

**Fig. 5 Factors involved in m⁶A regulated expression of PDK4. a** RIP-qPCR analysis of *PDK4* mRNA in wild type HeLa cells by use of antibody of YTHDF2, YTHDF3, and IGF2BP1-3. **b** IGF2BP3 RIP-qPCR analysis of *PDK4* mRNA in wild type or *Mettl3^Mut/-* HeLa cells. **c** Wild type or *Mettl3^Mut/-* HeLa cells were transfected with si-NC or si-IGF2BP3 for 24 h, the expression of PDK4 was checked by western blot analysis (left) and quantitatively analyzed (right). **d**, **e** HeLa cells were transfected with si-NC, si-IGF2BP3, pcDNA-PDK4-5'UTR-WT (**d**) or pcDNA-PDK4-5'UTR-Mut1 (**e**) for 24 h and then further treated with Act-D for the indicated times. The mRNA of *PDK4* was checked by qRT-PCR. **f** YTHDF1 RIP-qPCR analysis of *PDK4* mRNA in wild type or *Mettl3^Mut/-* HeLa cells. **g** Binding of YTHDF1 with the 5'UTR or 3'UTR in wild type or *Mettl3^Mut/-* HeLa cells were analyzed by YTHDF1 RIP-qPCR using fragmented RNA. **h** Wild type or *Mettl3^Mut/-* HeLa cells were transfected with vector or YTHDF1 construct for 24 h, the expression of PDK4 was checked by western blot analysis (left) and quantitatively analyzed (right). **i** HeLa cells were transfected with vector, YTHDF1 construct, pcDNA-PDK4-5'UTR-WT, and pcDNA-PDK4-5'UTR-Mut1 for 24 h, the expression of PDK4 was checked by western blot analysis (left) and quantitatively analyzed (right). **j** eEF-1 and eEF-2 RIP-qPCR analysis of *PDK4* mRNA in wild type or *Mettl3^Mut/-* HeLa cells. Data are presented as the mean ± SD from three independent experiments. *$p < 0.05$, **$p < 0.01$, NS, no significant, by two-tailed unpaired Student's *t* test for (**a**) ($p < 0.0001$) and (**g**) ($p < 0.0001$ and $p = 0.11$, respectively), and two-way ANOVA for (**b**) ($p < 0.0001$ and $p = 0.004$, respectively), (**c**, **d**) ($p < 0.0001$ for all), (**f**) ($p = 0.0003$ and $p = 0.001$, respectively), (**h–j**).

(Supplementary Fig. 6B) or gRNAs combined with dCas13b (Supplementary Fig. 6C) had no significant change.

The demethylation effect of dm⁶ACRISPR on PDK4 mRNA was verified by SELECT-qPCR. Results showed that the m⁶A levels of targeted site on PDK4 mRNA significantly decreased after transfecting cells with gRNAs and dCas13b-ALKBH5 (Fig. 6c). The strongest demethylation effect on PDK4 5'UTR was observed with gRNA1, which targets a ~90 nt downstream region from the m⁶A site and resulted in 80.2 ± 3.6% demethylation (2-ΔCt method). The dm⁶ACRISPR induced demethylation of PDK4 was further confirmed by m⁶A-RIP-qPCR (Supplementary Fig. 6D).

Results showed that dm⁶ACRISPR targeting PDK4 led to a significant downregulation of *PDK4* mRNA (Fig. 6d) and protein (Fig. 6e). This might be due to that dm⁶ACRISPR with gRNA for PDK4 can significantly decrease the binding of *PDK4* mRNA with YTHDF1 and IGF2BP3 (Fig. 6f). To investigate whether m⁶A mediated mRNA stability of PDK4 was related to the dm⁶ACRISPR induced upregulation of PDK4, we compared the effects of dCas13b-ALKBH5 with control gRNA or gRNA1 on

*PDK4* mRNA half-life. Results showed that targeted demethylation of PDK4 can significantly destabilize its mRNA (Fig. 6g), indicating that dm⁶ACRISPR decreased the mRNA stability via demethylation of m⁶A at 5'UTR in the case of PDK4.

To further investigate that dm⁶ACRISPR targeting PDK4 can modulate cell metabolism, we generated catalytically inactive dCas13b-ALKBH5 mutant (H204A)[6]. Our data showed that gRNA1 combined with dCas13b-ALKBH5 can significantly decrease the glucose consumption (Fig. 6h), lactate production (Fig. 6i), and ATP productions (Fig. 6j) as compared with that of non-targeted gRNA combined with dCas13b-ALKBH5. However, gRNA1 for PDK4 combined with dCas13b-ALKBH5 mutant had no significant effect on glycolysis and ATP generation (Fig. 6h–j). These data suggested that targeting m⁶A of PDK4 by dm⁶ACRISPR can regulate the glycolysis and ATP generation of cancer cells.

**TBP transcriptionally regulates expression of Mettl3.** We evaluated the potential mechanisms responsible for the expression

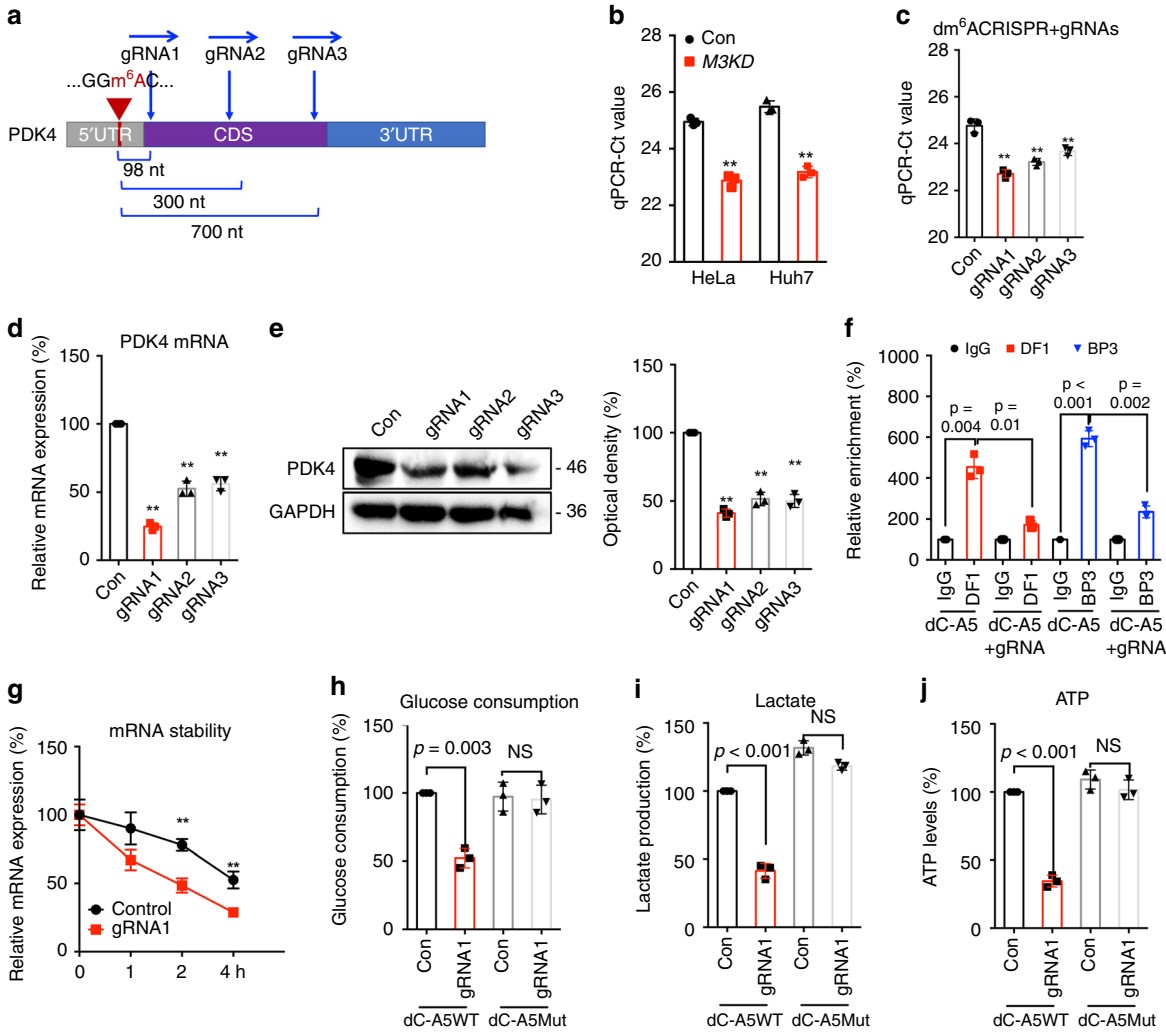

**Fig. 6 Targeting m⁶A of PDK4 by dm⁶ACRISPR re-programs metabolic of cancer cells. a** Schematic representation of positions of m⁶A site within *PDK4* mRNA and the regions targeted by three gRNAs, respectively. **b** The threshold cycle (Ct) of qPCR showing SELECT results for detecting m⁶A site in the potential m⁶A site of *PDK4* 5′UTR in *Mettl3*$^{Mut/-}$ HeLa, sh-Mettl3 Huh7 and their corresponding control cells. **c** The threshold cycle (Ct) of qPCR showing SELECT results for detecting m⁶A site in 5′UTR of *PDK4* in HeLa cells transfected with dCas13b-ALKBH5 combined with gRNA control or gRNA1/2/3, respectively, for 24 h. **d**, **e** The mRNA (**d**) or protein (**e**) expression of PDK4 in HeLa cells transfected with dCas13b-ALKBH5 combined with gRNA control or gRNA1/2/3, respectively, for 24 h. **f** RIP-qPCR analysis of *PDK4* mRNA in HeLa cells transfected with dCas13b-ALKBH5 combined with gRNA control (dC-A5) or gRNA for PDK4 (dC-A5 + gRNA) for 24 h by use of antibodies against YTHDF1 and IGF2BP3, respectively. **g** HeLa cells were transfected with gRNA control, gRNA1 for PDK4, and dCas13b-ALKBH5 for 24 h and then further treated with Act-D for the indicated times. The mRNA of *PDK4* was checked by qRT-PCR. **h–j** The glucose consumption (**h**), lactate production (**i**), and ATP levels (**j**) in HeLa cells transfected with gRNA control, gRNA1 for *PDK4*, and dCas13b-ALKBH5 or dCas13b-ALKBH5-Mut for 24 h. Data are presented as means ± SD from three independent experiments. **$p < 0.01$, NS, no significant, by two-tailed unpaired Student's $t$ test for (**b**) ($p = 0.001$ and $p = 0.002$, respectively), (**h–j**) one-way ANOVA for (**c**) ($p < 0.001$), (**d**) ($p < 0.001$), and (**e**) ($p < 0.001$), and two-way ANOVA for (**f**) and (**j**) ($p < 0.001$).

of Mettl3 in cancer cells. The expression of Mettl3 in ECT1/E6E7, a human normal cervical epithelium cell line, and cervical cancer HeLa and SiHa cells were checked. Our data showed that the protein (Fig. 7a) and mRNA (Fig. 7b) expression of Mettl3 increased in cervical cancer HeLa and SiHa cells as compared with that in ECT1/E6E7 cells. In addition, the precursor mRNA of Mettl3 in cervical cancer cells was also greater than that in ECT1/E6E7 cells (Fig. 7c). Further, we generated the promoter reporter of Mettl3 via inserting 1 Kb upstream of the transcription start site (TSS) of Mettl3 to pGL3 plasmid. The luciferase assay showed that the promoter activities of Mettl3 in cervical cancer cells were significantly greater than that in ECT1/E6E7 cells (Fig. 7d). It indicated that the transcription of Mettl3 was activated in cervical cancer cells.

We then evaluated the potential transcription factors (TFs) responsible for the regulation of Mettl3 in cancer cells. To identify TFs that directly regulate Mettl3 expression, we analyzed the ENCODE chromatin immunoprecipitation sequencing (ChIP-seq) data in ChIPBase[34] and PROMO with 5% maximum matrix dissimilarity rate[35]. Among the 47 factors identified by ChIPBase and 32 factors identified by PROMO, eight factors including ETS1, FOXA1, NRF1, PAX5, STAT4, TBP, TP53, and YY1 were overlapping between the two databases (Fig. 7e). We further compared the expression of these factors among ECT1/E6E7 and cervical cancer cells. Our data showed that only mRNA of NRF1 and TBP were upregulated in both HeLa and SiHa cells as compared with that in ECT1/E6E7 cells (Fig. 7f). Western blot analysis showed that the expression of TBP, rather than NRF1, in

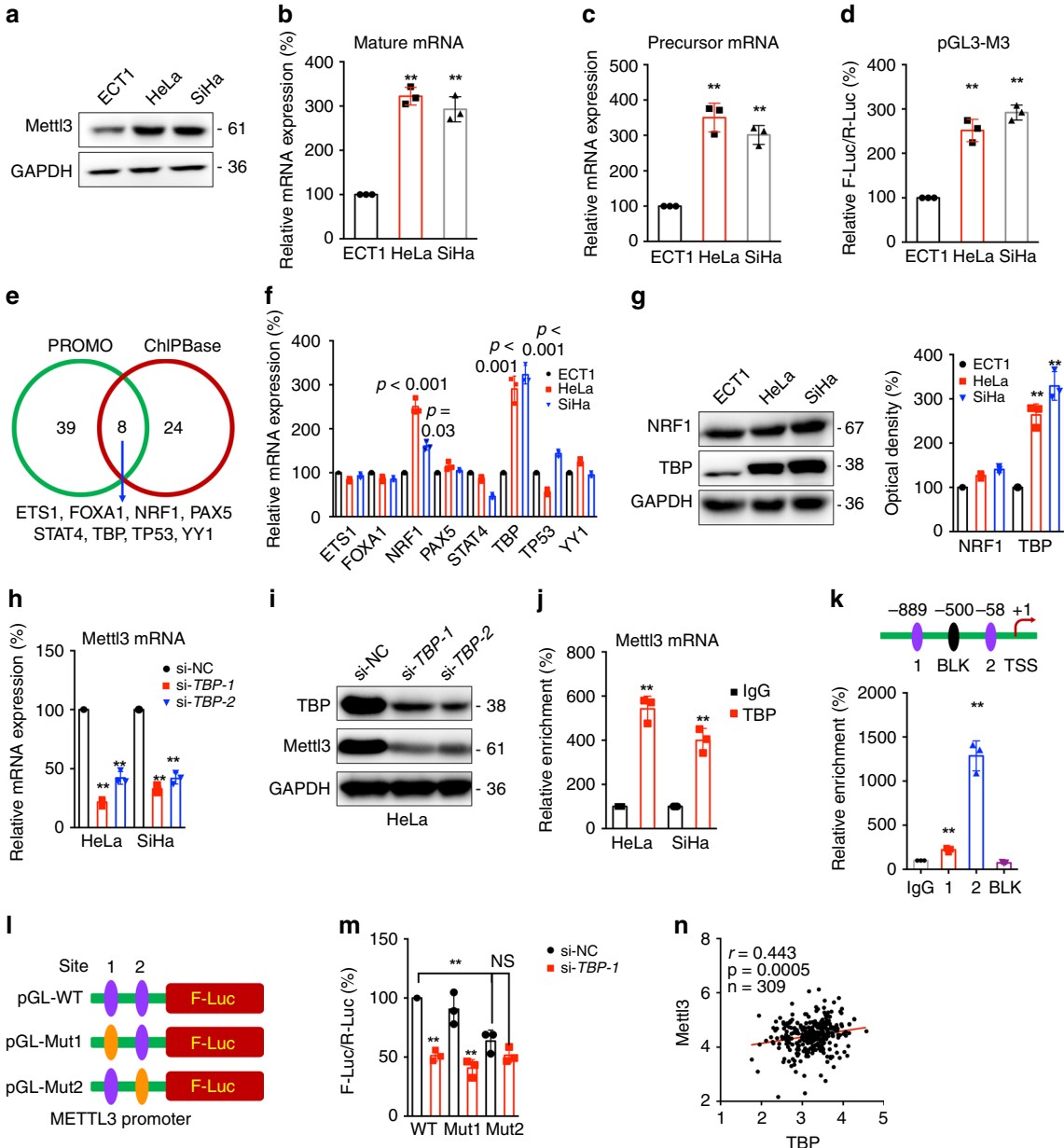

**Fig. 7 TBP is responsible for the upregulation of Mettl3 in cervical cancer cells. a–c** The protein (**a**), mature mRNA (**b**), or precursor mRNA (**c**) of Mettl3 in ECT1/E6E7, HeLa, and SiHa cells were checked. **d** The promoter activities of Mettl3 in ECT1/E6E7, HeLa, and SiHa cells were checked by dual-luciferase assay. **e** Venn diagram shows the overlap of transcription factors of Mettl3 predicted by PROMO and ChIPBase, respectively. **f** The mRNA expression of potential transcription factors of Mettl3 in ECT1/E6E7, HeLa, and SiHa cells were checked by qRT-PCR analysis. **g** The protein expression of NRF1 and TBP was checked by western blot analysis (*left*) and quantitatively analyzed (*right*). **h, i** Cells were transfected by siRNA negative control (si-NC) or siRNAs of TBP for 24 h, the mRNA (**h**) and protein (**i**) levels of Mettl3 were checked. **j** The binding between TBP and promoter of Mettl3 was checked by ChIP-PCR using IgG or TBP antibody. **k** Binding between TBP transcriptional factor and the promoter of Mettl3 at the potential binding site 1 and 2 or negative site BLK was checked by ChIP-PCR. **l** Schematic representation of the mutated promoter in pGL3-Basic-Mettl3-luc reporter to investigate the role of TBP in Mettl3 expression. **m** HeLa cells were co-transfected with pGL3-Mettl3-WT-Luc, pGL3-Mettl3-Mut1-Luc, pGL3-Mettl3-Mut2-Luc, pRL-TK plasmid and si-NC or si-TBP-1 for 24 h. Results were presented as the ratio between the activity of the reporter plasmid and pRL-TK. **n** Correlation between *Mettl3* and *TBP* in cervical cancer patients ($n = 309$) from ChIPBase database. Data are presented as the mean ± SD from three independent experiments. **$p < 0.01$. NS, no significant, by one-way ANOVA for (**b**) ($p < 0.001$ for all), (**c**) ($p < 0.001$ for all), (**d**) ($p < 0.001$ for all), (**f**, **g**) ($p < 0.001$ for all), and (**h**) ($p < 0.001$ for all), two-tailed unpaired Student's $t$ test for (**j**) ($p < 0.001$ for all), (**k**) ($p < 0.001$ for all), and (**n**) ($p = 0.0005$), and by two-way ANOVA for (**m**) ($p < 0.001$ for all).

cervical cancer cells was significantly greater than that in ECT1/E6E7 cells (Fig. 7g). We then knocked down the expression of TBP in HeLa and SiHa cells by use of its specific siRNA (Supplementary Fig. 7A). Our data showed that knockdown of TBP can suppress the mRNA (Fig. 7h) and protein (Fig. 7i and

Supplementary Fig. 7A) expression of Mettl3 in both HeLa and SiHa cells.

Chromatin immunoprecipitation (ChIP)-qPCR assays demonstrated that TBP had a significant enrichment of Mettl3 promoter over normal immunoglobulin G (IgG) control in both HeLa and

SiHa cells (Fig. 7j, indicating a direct binding between TBP and Mettl3 promoter. ChIPBase data (Supplementary Fig. 7B)[36] and JASPAR (Supplementary Fig. 7C) showed that there were two TBP potential binding sites in the promoter within 1 Kb upstream of Mettl3. ChIP-qPCR showed that binding of TBP to the potential binding site 2 was much greater than to the site 1 in HeLa cells (Fig. 7k). We then mutated the two TBP potential binding sites of promoter reporter of Mettl3 to generate the pGL-M3-Mut1 or pGL-M3-Mut2 (Supplementary Fig. 7D and Fig. 7l). Our data showed that si-TBP can significantly decrease luciferase levels of pGL-M3-WT and pGL-M3-Mut1, while the inhibition effect of si-TBP was attenuated for pGL-M3-Mut2 (Fig. 7m). Consistently, the relative values of F-Luc/R-luc of pGL-M3-WT and pGL-M3-Mut1 in HeLa cells were greater than that in ETC1/E6E7 cells, while this effect was attenuated for pGL-M3-Mut2 (Supplementary Fig. 7E). Further, the expression of TBP was significantly and positively correlated with the expression of Mettl3 in clinical cervical cancer patients from ChIPBase[34] (Fig. 7n), GEPIA (Supplementary Fig. 7F), or TCGA database (Supplementary Fig. 7G). All these data suggested that TBP might be responsible for the upregulation of Mettl3 in cervical cancer cells via binding to its promoter-proximal site to increase the transcription.

**PDK4 is involved in m6A regulated cancer progression.** Glycolysis and ATP generation are critical for cancer progression including growth, metastasis and chemoresistance[37,38]. We further over expressed PDK4 in $Mettl3^{Mut/-}$ HeLa, sh-Mettl3 Huh7, or si-Mettl3 SiHa cells (Supplementary Fig. 8A, B, C). Over expression of PDK4 can reverse the suppressed growth rate of Mettl3-deleted HeLa (Fig. 8a) and SiHa (Supplementary Fig. 8D) cells as compared to that of their corresponding control cells. Consistently, PDK4 also attenuated sh-Mettl3 suppressed proliferation of Huh7 cells (Fig. 8b). Further, our data suggested that deletion of Mettl3 increased sensitivity of HeLa cells to the treatment of doxorubicin (Dox), however, over expression of PDK4 can attenuate this effect and reduce the Dox sensitivity (Fig. 8c). It suggested that PDK4 was involved in Mettl3 regulated growth and chemosensitivity of cervical and liver cancer cells.

HeLa wide type, $Mettl3^{Mut/-}$, HeLa$^{PDK4}$, and $Mettl3^{Mut/-PDK4}$ stable cells were used to establish xenografts. Consistently, xenograft model confirmed that over expression of PDK4 can attenuate the suppression effect of $Mettl3^{Mut/-}$ HeLa cells on in vivo tumor growth (Fig. 8d). IHC showed that Mettl3 depletion led to a lower level of PDK4 in xenograft tumor tissues (Fig. 8e), which suggested that deletion of Mettl3 can regulate the expression of PDK4 in vivo.

At this point, we explored the possibility of a link between m6A methylation, PDK4 and cervical cancer development. The expression of Mettl3 was positively correlated with the PDK4 mRNA in cervical cancer patients (Fig. 8f). Further, Mettl3 expression in cervical cancer tissues was significantly ($p < 0.01$) greater than that in normal tissues, according to Zhai Cervix, Biewenga Cervix, and Pyeon Muti-Cancer data from the Oncomine database (Fig. 8g). Consistently, IGF2BP3 (Fig. 8h) and YTHDF1 (Fig. 8i), the readers of PDK4 mRNA, significantly increased in cervical cancer tissues as compared with that in the normal control samples. Using the online bioinformatics tools including GEPIA and Kaplan–Meier plotter, we found that cervical cancer patients with increased expression of Mettl3 (Fig. 8j) and TBP (Fig. 8k) showed reduced disease-free survival (DFS). Consistently, cervical cancer patients with increased expression of PDK4 showed reduced disease-free survival (DFS, Fig. 8l) and overall survival (OS, Fig. 8m). Together, these data

suggested that PDK4 was involved in m6A regulated glycolysis and progression of cervical cancer.

## Discussion

The metabolic alterations are generally thought to promote cancer progression[39]. Our data showed that m6A can direct carbon flux into glycolysis from OXPHOS. A few studies suggested that potential clues between RNA methylation and lipid metabolism[17,40,41]. For example, m6A negatively mediated UCP2 protein expression and positively mediated PNPLA2 protein expression to regulate obesity development[41]. Knockdown of Mettl3 increases PPaRα mRNA lifetime and expression, which resulted in the reducing lipid accumulation and cell viability[17]. Our data found that deletion of Mettl3 or over expression of ALKBH5 can suppress glucose consumption, lactate production, ATP generation, and ECAR. Further, GO and GSEA analysis revealed that deletion of Mettl3 downregulated genes enriched in several key ontology terms reflecting cellular metabolic processes such as glycolysis and glycan degradation. Our results described the potential roles of m6A in cancer metabolism, and also created the possibility to develop therapeutic strategies against cancer progression by targeting m6A modification.

The conventional Warburg effect is generally associated with upregulation of PDK1/2/3[42], whereas the expression of these enzymes is not varied in m6A regulated Warburg-effect-like metabolic changes; instead, PDK4, which directs carbon flux into glycolysis from OXPHOS[20], acts as the key upstream regulator. PDK4 is the most widely distributed PDK isoform which plays oncogenic roles in human colon[25] and bladder[43] cancer. Our data showed significant enrichment of m6A in its 5′UTR and 3′UTR regions of PDK4 in HeLa cells. Deletion of Mettl3 decreased the expression of PDK4, while over expression of PDK4 can attenuate Mettl3 regulated glycolysis of cancer cells. It confirmed the essential roles of PDK4 in m6A regulated glycolysis and ATP generation.

The m6A modification can regulate nearly all stages in the life cycle of RNA, such as RNA processing, nuclear export and translation modulation[2,44]. Our data showed that m6A in 5′UTR, rather than 3′UTR, positively regulated the mRNA stability and translation of PDK4. YTHDF1-eEF-2 interaction is likely involved in the m6A regulated translation elongation of PDK4 mRNA in cancer cells, while IGF2BP3 was involved in m6A regulated mRNA stability. Consistently, our recent study indicated that methylation of CDS of Snail, which recruited the YTHDF1 and eEF-2, can trigger its translation elongation and cancer metastasis[12]. IGF2BPs can bind the GG(m6A)C sequence of mRNA to promote the stability and storage of their target mRNAs[26]. The oncogenic roles of IGF2BP3 have been observed in various cancers[45]. Our data provided a new insight into the function of IGF2BP3 and YTHDF1 in glycolysis via regulating the mRNA stability and translation of PDK4, respectively.

We further specifically demethylated the m6A of PDK4 mRNA by use of dm6ACRISPR system[32]. The system resulted in about 80% of demethylation and significantly decreased the expression of PDK4 and glycolysis of cancer cells. dm6ACRISPR is a newly developed method that targeting demethylation of specific mRNA in transcriptome to artificially manipulate cell metabolism. Further, both in vitro and in vivo data suggested that PDK4 was involved in Mettl3 regulated growth and chemosensitivity of cancer cells. Clinical analysis confirmed the positive correlation between Mettl3 and PDK4 in cervical cancer tissues. Both high expression of PDK4 and Mettl3 reduced the survival rate of cancer patients. Mettl3 has been reported to maintain myeloid leukemia[46], promote liver cancer progression[47], and is essential for GSC maintenance and radioresistance[48]. Considering that

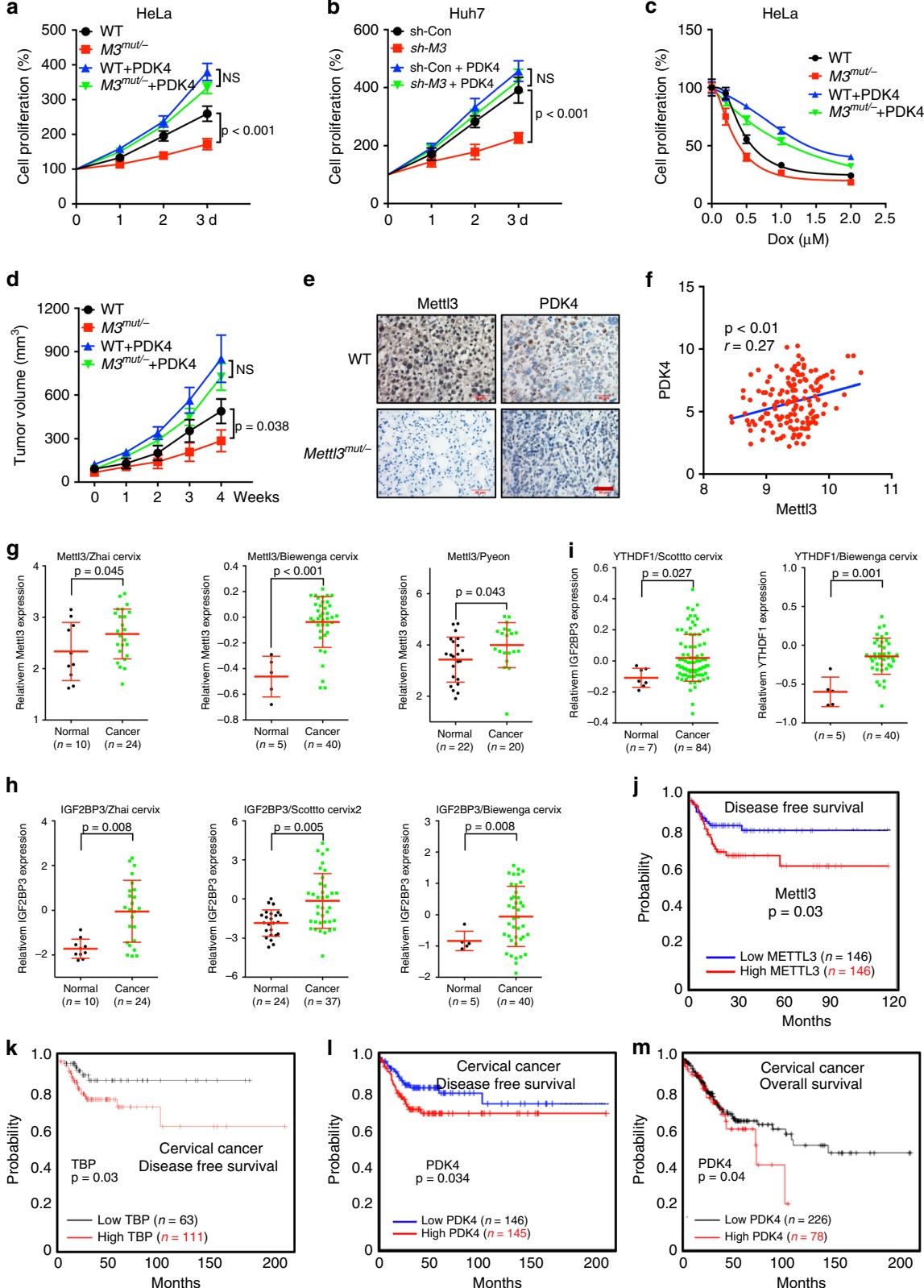

metabolic control by PDK4 is critical for cancer development, the roles of PDK4 in m6A related oncogenic functions in other cancers need further investigation.

Finally, we found that TBP can regulate the transcription of Mettl3 to increase its expression in cervical cancer cells. Nowadays, TFs which can regulate the expression of RNA methyltransferases are not well illustrated. It has been reported that the expression of Mettl14 is negatively regulated by SPI1 in acute myeloid leukemia (AML) cells[49]. As to Mettl3, hepatitis B X-interacting protein (HBXIP) can increase its expression in breast cancer cells via inhibiting miRNA let-7g, which downregulated the expression of Mettl3 by targeting its 3′UTR[50]. MiR-33a can decrease the

**Fig. 8 PDK4 is involved in m⁶A regulated cancer progression.** The relative cell proliferation of wild type and *Mettl3*^{Mut/-} HeLa cells stably transfected with vector control or PDK4 constructs; The relative cell proliferation of sh-control and sh-*Mettl3* Huh7 cells stably transfected with vector control or PDK4 constructs; Wild type and *Mettl3*^{Mut/-} HeLa cells stably transfected with vector control or PDK4 constructs were treated with increasing concentrations of Dox for 24 h, and the cell proliferation was tested. **a** The tumor growth curves of wild type and *Mettl3*^{Mut/-} HeLa cells stably transfected with vector control or PDK4 constructs. **b** IHC (Mettl3 and PDK4)-stained paraffin-embedded sections obtained from wild type and *Mettl3*^{Mut/-} HeLa cells. The scale bar is 50 μM. **c** Correlation between *Mettl3* and *PDK4* in cervical cancer patients (*n* = 169) from TCGA database. **g–i** The relative mRNA expression of *Mettl3* (G), *IGF2BP3* (H), or *YTHDF1* in Oncomine datasets. **j, k** The Kaplan–Meier survival curves of DFS based on Mettl3 (**j**) or TBP (**k**) expression in cervical cancer patients from TCGA data base. **l, m** The Kaplan–Meier survival curves of DFS (**l**) or OS (**m**) based on PDK4 expression in cervical cancer patients from TCGA data base. Data are presented as means ± SD from three independent experiments. A representative from a total of three independent experiments is shown for (**e**). NS, no significant, by one-way ANOVA for (**a**), (**b**), and (**d**), by two-tailed unpaired Student's *t* test for (**f–i**), and by two-sided log-ranjk test for (**j–m**).

expression of Mettl3 in lung cancer cells to suppress cell proliferation[51]. In gastric cancer cells, H3K27ac can activate Mettl3 transcription to increase its expression[52]. Our data provided a novel mechanism which was responsible for the upregulation of Mettl3 in cancer cells.

In summary, we provided compelling in vitro and in vivo evidences demonstrating that m⁶A can regulate the glycolysis of cancer cells via regulation of PDK4. The methylation of 5′UTR is critical for m⁶A mediated mRNA stability and translation. Since a great number of genes are involved in cell metabolism, we cannot exclude the possibility that m⁶A modification regulates metabolic process by indirectly targeting other genes. Our study suggested that m⁶A regulates glycolysis of cervical and liver cancer cells through induction of PDK4, which expanded our understanding of such interplays that are essential for therapeutic application.

## Methods

**Metabolic assay**. The glucose and lactate concentration in cultured media were measured using commercial kits (BioVision) following the manufacturer's instructions. ATP was quantified using Cell Titer-Glo® luminescence assay (Promega) according to the manufacturer's instructions. All samples were tested in triplicate.

**ECAR and OCR**. Extracellular acidification rate (ECAR) and oxygen consumption rate (OCR) were analyzed using the Seahorse XF96 instruments (Seahorse Bioscience, USA). For OCR test, cell medium was replaced by assay medium (Seahorse Bioscience) supplemented with 1 mM pyruvate, 10 mM glucose, and 2 mM glutamine for 1.5 h at 37 °C, then measured by the XF Cell Mito Stress Kit (Seahorse Bioscience). The concentrations of oligomycin and FCCP were 1.0 μM and 0.5 μM, respectively. For ECAR test, cells were incubated in the assay medium (Seahorse Bioscience) with 2 mM glutamine for 1.5 h at 37 °C, then measured by the Glycolytic Stress Test Kit (Seahorse Bioscience). The OCR and ECAR results were adjusted to the Seahorse XF96 Wave software.

**RNA-seq and bioinformatics**. Total RNAs from wild type and *Mettl3*^{Mut/-} HeLa cells were purified using RNeasy mini kit (Qiagen, Hilden, Germany). Then the cDNA was generated by use of a NuGEN Ovation RNA-Seq Systemv2 (NuGEN, San Carlos, CA). As to mRNA-seq, total mRNA preparation and sequencing were performed by the Beijing Genomics Institute (BGI, Shenzhen, Guangdong, China). Sequencing reads were mapped to reference human genome sequence (NCBI 36.1 [hg19] assembly by TopHat (Version 2.0.6). The read counts were expressed as FPKM. The differential expression between conditions was statistically assessed by R/Bioconductor package edgeR (version 3.0.8). Genes with FDR of <0.05 and >200 bp were called as differentially expressed.

For GESA, standard procedure (http://www.broadinstitute.org/gsea/doc/GSEAUserGuideFrame.html) as described by GSEA user guide was performed. The curated gene set C2 of the Molecular Signature Database version 4.0 was used to compute overlaps between gene sets in Molecular Signature Database and our gene set.

**Cell line and cell culture**. Human cancer HeLa, SiHa, Huh7, HepG2, and MDA-MB-231 cells were purchased from the American Type Culture Collection (ATCC, Manassas, VA) and cultured in Dulbecco's modified Eagle medium (DMEM, GIBCO, Carlsbad, CA, USA) with 10% fetal bovine serum (FBS) and 1 % penicillin/streptomycin (Invitrogen). The ECT1/E6E7, a human normal cervical epithelium cell line, was kindly provided by Prof Tianfeng Chen at Jinan University. The ECT1/E6E7 cells were grown in keratinocyte-serum free medium (GIBCO)

containing 0.1 ng/ml human recombinant EGF, 0.05 mg/ml bovine pituitary extract, and additional calcium chloride 44.1 mg/L.

**Plasmid, siRNA, shRNA and generation of stable cell lines**. The CDS of Mettl3, ALKBH5 and PDK4 were cloned into pcDNA3.1 to generate over expression plasmid. pcDNA3.1 was used as the vector control for analysis. Mettl3 mutant DA (D395A) and ALKBH5 mutant (H204A) plasmids were generated in our previous study[53]. For IGF2BP3, Mettl3, and TBP knockdown, three synthesized duplex RNAi oligos targeting human mRNA sequences from Sigma were used. A scrambled duplex RNA oligo (5′-UUCUCCGAACGUGUCACGU) was used as RNA control. Twenty-four hours before transfection, the medium was replaced with fresh medium and transfected using Lipofectamine 2000 reagent (Invitrogen) with vector control, plasmid construct, siRNA negative control (si-NC), or siRNAs according to the manufacturer's instructions. The working concentration of siRNA was 50 nM.

The *Mettl3*^{Mut/-} HeLa cells were generated by CRISPR-cas9 editing system in the previous study[54]. The sh-control and sh-*Mettl3* Huh7 cells were kindly provided by Prof Jun Cui at Sun Yat-sen University. To generate PDK4 stable over expression cells, the wide type and *Mettl3*^{Mut/-} HeLa cells were transfected with pcDNA/PDK4 or vector control via liposome-mediated transfection. G418 (700 μg/ml) was used to select the transfected cells. After two week selection, the survived cells were seeded individually into 96-well plate for further expansion. The expression of PDK4 and Mettl3 was checked by western blot analysis. For over expression of target genes, cells were transfected with plasmids by lipofectamine 2000 according to the manufacturers' instructions.

To generate dCas13b-ALKBH5 fusion protein, the original PspCas13b plasmid (Addgene plasmid #103866), gRNA plasmid (Addgene plasmid #103854) and non-targeting gRNA plasmid (Addgene plasmid #103868) were obtained from Addgene. PspCas13b-Alkbh5 and gRNA-containing plasmids were constructed by Synbio Technologies Company (Suzhou, China). Inactive Cas13b was constructed containing double mutations at A133H and A1058H of PspCas13b.

**Luciferase reporter assay**. To evaluate the effect of 3′UTR on PDK4 expression, the wild type or mutant-1/-2/-3 of 3′UTR of *PDK4* was inserted behind the F-luc coding region. Both the pmirGLO-PDK4-3′UTR-WT and pmirGLO-PDK4-3′UTR-Mut-1/-2/-3 were transfected into wild type or *Mettl3*^{Mut/-} cells for 24 h, the firefly luciferase (F-luc) and renilla luciferase (R-luc) were assayed by Dual-Glo Luciferase Assay system (Promega).

To evaluate the potential roles of 5′UTR in PDK4 expression, the wild type or mutant 5′UTR ligated with promoter of PDK4 was inserted in the front of F-luc coding region of pGL3-basic plasmid to generate pGL-3-PDK4-5′UTR-WT, pGL-3-PDK4-5′UTR-Mut1, and pGL-3-PDK4-5′UTR-Mut2, respectively. Both these plasmids were co-transfected with pRL-TK into wild type or *Mettl3*^{Mut/-} cells for 24 h. The firefly luciferase (F-luc) and renilla luciferase (R-luc) were assayed by Dual-Glo Luciferase Assay system (Promega). Renilla Luciferase (R-luc) was used to normalize firefly luciferase (F-luc) activity. The translation outcome was determined as a relative signal of F-luc divided by R-luc, the mRNA abundance was determined by qRT-PCR of F-luc and R-luc, and the translation efficiency of PDK4 is defined as the quotient of reporter protein production (F-luc/R-luc) divided by mRNA abundance[28].

Promoter activity of Mettl3 and PDK4 in cells was measured by luciferase assay according to our previously described protocol[55]. Briefly, cells were transfected with pGL3-Mettl3-WT-Luc, pGL3-Mettl3-Mut1-Luc, or pGL3-Mettl3-Mut2-Luc containing the −1000/+1 sequence of the Mettl3 promoter, or pGL3-Basic-PDK4-luc containing sequence −1000/+1 of PDK4 promoter.

Transfection efficiency was normalized by co-transfection with pRL-TK. After 24 h incubation, luciferase activity was measured using the Dual Luciferase Reporter Assay kit (Promega) according to the manufacturer's instructions. Renilla Luciferase (R-luc) was used to normalize firefly luciferase (F-luc) activity to evaluate the reporter transcription. Experiments were performed three times with similar results.

**mRNA stability**. Stability of RNA in wild type and *Mettl3^Mut/-* HeLa cells was achieved by incubating cells with actinomycin D (Act-D, Catalog #A9415, Sigma, U.S.A) at 5 µg/ml. Cells were then collected at the indicated times and RNA was isolated for Real-time PCR. Half-life ($t_{1/2}$) of *PDK4 mRNA* was calculated using ln2/ slope and GAPDH was used for normalization.

**Polysome profiling**. The fraction of ribosome was separated by centrifugation in a sucrose gradient. Cells pretreated with 100 µg/ml cycloheximide were lysed in 1 ml lysis buffer (10 mM Tris, pH 7.4, 150 mM KCl, 5 mM MgCl2, 100 µg ml⁻¹ CHX, 0.5% Triton-X-100, freshly add 1:100 protease inhibitor, 40 U ml⁻¹ SUPERasin). After centrifugation at 15,000 g for 15 min, the supernatant was separated by 5/50% w/v sucrose gradient at 4 °C for 4 h at 140, 000 g (Beckman, rotor SW28). The sample was then fractionated and analyzed by Gradient Station (BioCamp) equipped with ECONO UV monitor (BioRad) and fraction collector (FC203B, Gilson). The fractions resulting from sucrose gradient were used for RNA extraction and qRT-PCR.

**Design of the guide RNAs**. mRNA sequences of all isoforms of target genes were subjected to alignment analysis to identify the common regions, which were acted as targeting candidates for gRNA design. gRNAs targeting CDS region of PDK4 were designed, all designed gRNAs were subject to MEGABLAST (https://blast.ncbi.nlm.nih.gov/Blast.cgi) to avoid mismatching to unexpected mRNA in human genome. The sequences of gRNAs were: gRNA1, 5′-AGA ATG TTG GCG AGT CTC AC-3′; gRNA2, 5′-GGA TCA ATG CTT CCA ATG TGG CTT GGG TTT CC-3′; gRNA3, 5′-CAG TTA GGA TCA ATG CTT CCA ATG TGG CTT GG-3′.

**SELECT qPCR**. SELECT qPCR method was following Xiao's protocol[33] with slight modifications. Qubit (Thermo Fisher Scientific) with Qubit™ RNA HS Assay Kit (Thermo Fisher Scientific) was used to quantified total RNAs. Then, 1500 ng of total RNA was mixed with 40 nM up and down primers and 5 µM dNTP in 17 µl 1 x CutSmart buffer (NEB). The mixture was incubated with the follow program: 90 °C for 1 min, 80 °C for 1 min, 70 °C for 1 min, 60 °C for 1 min, 50 °C for 1 min and 40 °C for 6 min. The sample was further mixed with 0.5 U SplintR ligase, 10 nM ATP and 3 µl of 0.01 U Bst 2.0 DNA polymerase and incubated at 40 °C for 20 min and denatured at 80 °C for 20 min. Afterwards, 20 µl qPCR reaction containing 2 µl of final reaction mixture, 2 x SYBR Green Master Mix (TaKaRa), and 200 nM SELECT primers (listed in Supplementary Table 3) was performed. The qPCR program was 95 °C, 5 min; (95 °C, 10 s; 60 °C, 35 s) × 40 cycles; 95 °C, 15 s; 60 °C, 1 min; 95 °C, 15 s; 4 °C, hold. Results were calculated by normalized the $C_t$ values of samples to their corresponding $C_t$ values of control. All assays were performed with three independent experiments.

**Xenograft models**. Female BALB/c nude mice (four weeks old) were purchased from Sun Yat-sen University (Guangzhou, China) Animal Center and raised under pathogen-free conditions. The animal experiments were complied with Zhongshan School of Medicine Policy on Care and Use of Laboratory Animals. As to the subcutaneous transplanted model, WT^Vec, *Mettl3^Mut/-+vec*, WT^PDK4, *Mettl3^Mut/--+PDK4* cells (2 × 10⁶ per mouse, n = 10 for each group) were diluted in 200 µL normal medium + 200 µL Matrigel (BD Biosciences) and subcutaneously injected into immunodeficient mice to investigate tumor growth. Tumor growth was monitored every three days. The tumor volume was calculated using the formula V = 1/2× larger diameter × (smaller diameter)².

**Database (DB) analysis**. Kaplan-Meier plotter (http://kmplot.com/analysis/)[56] was used to assess the prognostic value of Mettl3 and PDK4 expression in patients with cervical cancers. Expression levels of PDK4 in cancer tissues and normal tissues of cervical cancers were obtained from TCGA (The Cancer Genome Atlas) database. The correlation between Mettl3 and PDK4 was evaluated by use of LinkedOmics (http://www.linkedomics.org), which is a publicly available portal that includes multi-omics data from all 32 cancer types from TCGA[57]. PDK4 expression in cervical cancer from Oncomine database (Zhai Cervix, Biewenga Cervical, and Pyeon Mutiple Cancer) was analyzed. TF binding data and gene co-expression analysis were derived from ChIPBase 2.0[34].

**Western blot analysis**. Western blot analysis was performed as our previous study[58]. Briefly, cells were lysed by use of lysis buffer containing 50 mM Tris-HCl (pH 7.6), 150 mM NaCl, 1 mM EDTA, 1%NP-40, 0.5%Na-deoxycholate, 5 mg/ml aprotinin, 5 mg/ml leupeptin, and 1 mM phenylmethylsulfonyl fluoride for 30 minutes after treatment as indicated in figure legends. About 20 µg total proteins were separated on sodium dodecyl sulfate-polyacrylamide gel electrophoresis (SDS-PAGE), transferred onto polyvinylidene fluoride (PVDF) membranes, blocked with 5% non-fat milk at room temperature for 2 h, and incubated with primary antibodies. After washed three times, membrane was incubated with horseradish peroxidase (HRP)-conjugated secondary antibodies (1: 5,000 dilution) for 2 h at room temperature. The blot was visualized by using Western Blotting Plus Chemiluminescence Reagent (Life Science). The bands resulting from gel imaging were quantified by ImageJ software (National Institutes of Health, Bethesda, MD, USA). The antibodies used in the present study were: PDK4 (Abcam, ab89295, 1:500), Mettl3 (Abcam, ab195352,

1:500), ALKBH5 (Abcam, ab69325, 1:500), H2A.X (Abcam, ab26350, 1:500), IGF2BP3 (Santa Cruz, sc-365641, 1:500), YTHDF1(Abcam, ab99080, 1:500), NRF1 (Abcam, ab34682, 1:500), TBP (Abcam, ab818, 1:500) and GAPDH (BOSTER, BM3876, 1:1000). The bands resulting from gel imaging were quantified by ImageJ software (National Institutes of Health, Bethesda, MD, USA).

**RNA extraction and real-time PCR for gene expression**. Total RNA was isolated from target cells using Trizol reagent (Invitrogen) according to the manufacturer's recommendation. Then 500 ng of mRNA were used to synthesize the cDNA by using Prime Script RT Master Kit (Takara, Dalian, China). The qRT-PCR was performed on CFX96 Touch real time System (Biorad, USA) with 2 x SYBR Green Master Mix (TaKaRa). Primers of targeted genes were listed in Supplementary Table 3. The cycle number at threshold was used to quantify the transcript levels of genes of interest. Transcripts of the housekeeping gene GAPDH in the same incubations were used for normalization, while nuclear MALAT1 RNA was selected as endogenous control for the nuclear RNA. The relative expression levels were calculated by the 2-ΔCt methods.

**m⁶A-RIP qPCR**. 1 µg m⁶A or IgG antibody were incubated with Protein G Magnetic beads in 1x Reaction buffer (150 mM NaCl, 10 mM Tris-HCl, pH 7.5, 0.1% NP-40 in nuclease free $H_2O$) at 4 °C for 3 h, followed by incubation with 200 µg extracted RNA at 4 °C for 3 h. Incubation of RNA-antibody-conjugated beads with 100 µl Elution Buffer (75 nM NaCl, 50 nM Tris-HCl, pH 7.5, 6.25 nM EDTA, 1% (w/v) SDS, 20 mg/ml Proteinase K) for 30 min at room temperature was used to elute the bound RNAs. The eluted RNA was extracted by phenol: chloroform method followed by ethanol precipitation. Isolated m⁶A-RIP RNA was reverse transcribed and quantification by qPCR. IP enrichment ratio of a transcript was calculated as the ratio of its amount in IP to that in the input yielded from same amounts of cells.

**m⁶A sequencing (m⁶A-seq) and data analysis**. Total polyadenylated RNA was isolated from HeLa cells by use of TRIZOL reagent followed by isolation through FastTrack MAGMaxi mRNA isolation kit (Invitrogen). RNA fragmentation, m⁶A-seq, and library preparation were conducted according to manufacturer's instructions and previously published protocol[28]. The library was prepared by use of NEBNext Ultra Directional RNA Library Prep Kit (New England BioLabs, Ipswich, MA). Each experiment was conducted with two biological replicates. m⁶A-seq data was analyzed according to protocols described before[28]. Significant peaks with FDR < 0.05 were annotated to RefSeq database (hg19). Homer was used to identify the sequence motifs. Cufflinks using sequencing reads from input samples was used to calculate gene expression. Cuffdiff was used to find DE genes.

**Mitochondrial DNA assay**. Mitochondrial DNA (mtDNA) copy number was determined by a standard by real-time PCR using nuclear DNA (nDNA, GAPDH primer set) content as an internal control for normalization. Extract genomic DNA (including mtDNA) from target cells were isolated using the Qiagen Genomic DNA kit according to the manufacturer's instructions. Cells relative content of mtDNA (mtDNA primer set) was checked by qRT-PCR and normalized to that of nuclear DNA (GAPDH primer set). The primers were as follow: mtDNA, forward 5′-ACG CCA TAA AAC TCT TCA CCA AAG-3′, reverse 5′-GGG TTC ATA GTA GAA GAG CGA TGG-3′; GAPDH, forward 5′-ACA ACT TTG GTA TCG TGG AAG G-3′, reverse 5′-GCC ATC ACG CCA CAG TTT C-3′.

**LC-MS/MS assay for m⁶A quantification**. After purified using oligo dT magnetic beads, mRNA was digested with nuclease P1 (0.5 U, Sigma) in 25 µl reaction system containing 10 mM $NH_4OAc$ (pH = 5.3) at 42 °C for 1 h. Then mixtures were further incubated with $NH_4HCO_3$ (1 M, 3 µL) and alkaline phosphatase (1 µL, 1 U/µL; Sigma) at 37 °C for 2 h. After neutralization and filter, samples were separated by a C18 column (Agilent) and analyzed by an Agilent 6410 QQQ triple-quadrupole LC mass spectrometer using positive electrospray ionization mode. Ratio of m⁶A to A was calculated based on calibration curves.

**Cell proliferation assay**. Cells were seeded in 96-well plates at 1×10⁴ cells per well in 10% FBS-supplemented medium. After treatment as indicated conditions, cell proliferation was evaluated using CCK-8 cell viability assay system according to the manufacturer's protocol. A microplate reader was used to measure the absorbance at 450 nm.

**Protein stability**. Protein stability of targets in wild type and *Mettl3^Mut/-* HeLa cells were achieved by incubation of cycloheximide (CHX, final concentration 100 µg/ml) during indicated times. The expression of PDK4 was measured through western blot analysis.

**Subcellular fraction**. The nucleus and cytoplasm were separated by NE-PER nuclear and cytoplasmic extraction reagents (Pierce, Rockford, IL, USA). The fractions were pooled to isolate total RNA by TRIzol reagent for qRT-PCR.

GAPDH were used as endogenous control for cytoplasmic RNA, while MALAT1 RNA was selected as endogenous control for nuclear RNA.

**RIP-RT-PCR**. Before sample collection, cells were irradiated twice with 400 mJ/cm$^2$ at 254 nm by Stratalinker on ice. Cells were then lysed in high salt lysis buffer (300 mM NaCl, 0.2% NP-40, 20 mM Tris-HCl PH 7.6, 0.5 mM DTT, protease inhibitor cocktail (1 tablet/50 ml), and RNase inhibitor (1:200)) at 4 °C for 30 min. After treatment with or without 1 U RNase T1 for 15 min at 24 °C, 10% supernatant was collected as input. YTHDF1/2/3, IGF2BP1/2/3, or eEF-1/2 antibody, or IgG-conjugated Protein A/G Magnetic Beads were used to incubate the remaining supernatant in 1x IP buffer supplemented with RNase inhibitors at 4 °C overnight. 100 μl Elution Buffer (5 mM Tris-HCL pH 7.5, 1 mM EDTA pH 8.0, 0.05% SDS, 20 mg/ml Proteinase K) was used to elute the immunoprecipitated RNAs for 2 h at 50 °C. RNA was further isolated by phenol: chloroform method followed by ethanol precipitation. IP enrichment ratio of a transcript was calculated as ratio of its amount in IP to that in the input, yielded from same amounts of cells.

**Immunohistochemistry (IHC)**. All of slides were placed in a 60 °C incubator for 20 min, de-paraffinized in xylene and rehydrated in gradient ethanol. The slides were incubated with 3% hydrogen peroxide for 10 min, followed by antigen retrieving using 0.01 M citrate buffer (pH 6.0) for 30 min. After blocking with 5% BSA, slides were incubated overnight at 4 °C with relevant primary antibody. After that, the secondary biotin-conjugated antibody was applied for 1 h in room temperature. The IHC staining was visualized using diaminobenzidine reaction, counterstained with hematoxylin.

**Statistical analyses**. Data were reported as mean ± SD from at least three independent experiments. For statistical analysis, two-tailed unpaired Student's *t*-test between two groups and by one-way or two-way ANOVA followed by Bonferroni test for multiple comparison were performed. All statistical tests were two-sided. Data analysis was carried out using SPSS 16.0 for Windows. A *p*-value of < 0.05 was considered to be statistically significant. *$p < 0.05$, **$p < 0.01$; NS, no significant.

**URLs and data resources**. Gene Ontology Analysis (GO), http://pantherdb.org/; Gene Set Enrichment Analysis (GSEA), http://software.broadinstitute.org/gsea/msigdb/; NCBI Gene Expression Omnibus (GEO), http://www.ncbi.nlm.nih.gov/geo/.

**Consent**. The authors confirmed that we have obtained written consent from the patient to publish the manuscript.

**Reporting summary**. Further information on research design is available in the Nature Research Reporting Summary linked to this article.

## Data availability

The accession number for the high-throughput of m6A-seq data reported in this paper is GEO: GSE112795 (https://www.ncbi.nlm.nih.gov/geo/query/acc.cgi?acc=GSE112795). The data underlying Figs. 7n, and 8f–n referenced during the study are available in a public repository from the TCGA website. The source data underlying all experimental results (Figs. 2g, h; 3e, f left panel; 4h left panel; 5c, h, i left panel; 6e left panel; 7g left panel, i and Supplementary Figs. 1a left panel, d; 2d, f; 3a; 5b left panel; 7a; 8a-c) are provided as a Source Data file. All the other data supporting the findings of this study are available within the article and its Supplementary Information files. A reporting summary for this article is available as a Supplementary Information file.

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

## Acknowledgements

The authors thank Dr Feng Liu and Prof Junjiu Huang at School of Life Sciences, Sun Yat-sen University for technical support. This research was supported by the National Natural Science Foundation of China (Grant Nos. 81973343, 81673454, 81672608, and 31801197), the Guangdong Provincial Key Laboratory of Chiral Molecule and Drug Discovery (2019B030301005), the Guangdong Provincial Key Laboratory of Construction Foundation (No. 2017B030314030), the Fundamental Research Funds for the Central Universities (Sun Yat-sen University) (Nos.19ykpy130 and 19ykzd24), the Natural Science Foundation of Guangdong Province of China (No. 2020A1515010291), and the China Postdoctoral Science Foundation (No. 2018M643354).

## Author contributions

Conception and design: Hongsheng Wang, Zhuojia Chen; Acquisition of data: Zihan Li, Yanxi Peng, Jiexin Li, Zhuojia Chen, Feng Chen; Analysis and interpretation of data: Hongsheng Wang, Zihan Li, Yanxi Peng, Feng Chen, Jian Tu; Writing, review, and/or revision of the manuscript: Hongsheng Wang, Zhuojia Chen, Shuibin Lin.

## Competing interests

The authors declare no competing interests.
