## [Peer Review File · Nature Communications]

Reviewers' comments:

Reviewer #1 (Remarks to the Author):

In this study, authors show the strong correlation between RNA m6A modification and cell metabolism. Both Mettl3 knockdown and ALKBH5 overexpression resulted in the decreased the glucose metabolism, lactate production and ATP levels. The only one target, PDK4, was identified from Mettl3 deleted Hela cells. The m6A sites in the 5'-UTR regulated PDK4 mRNA stability and translation by binding with IGF2BP3 and YTHDF1/eEF2 complex, respectively. Further, The m6A in the 5'-UTR was demethylated with dm6ACRISPR developed by their lab. This experiment confirmed the 6mA regulated PDK4 expression affected the glycolysis of cancer cells. This work is novel and important for understanding the functional mechanism of mRNA m6A modification in cancer metabolism. For current version, the manuscript is well organized. However, the authors need to provide more details and stronger evidence to support their conclusion.

Below are the reviewer's comments:

- 1 In Figure 2C, author claimed that the only candidate was identified from m6A-seq and mRNA-seq data. It was not clear that how the data was analyzed, which software, what is the cut-off?
- 2 In Figure3, PDK4 mRNA stability and translation was decreased in mettl3 deleted cells. Did authors observe this result in ALKBH5 overexpression cells?
- 3 How many m6A sites in the 5'-UTR of PDK4? In the targeted demethylation assay, did authors check the binding of YTHDF1 and IGF2BP3?
- 4 In vivo assays, I think the glycolysis parameters should be tested.
- 5 Most experiments were performed in the Hela and Huh7 cell lines. They are derived from cervical cancer cells and liver cancer cells. The relationship of Mettl3 expression level and disease survival is not consistent in different cancers. It is better to narrow the concept scope. Please refer to <https://www.proteinatlas.org/ENSG00000165819-METTL3/pathology>
- 6 In Figure 6J, What does the curve means ? Is it Mettl3 with low PDK4 or High PDK4? Whether the two genes have synergic effect on the disease free survival?

Reviewer #2 (Remarks to the Author):

Major claims of the paper:

The authors reported findings using HeLa cells that mitochondrial PDK4 is positively regulated by m6A modified 5'-UTR, leading to enhanced translation elongation and mRNA stability through binding with m6A "readers" including YTHDF1/eEF-2 complex and IGF2BP3. The authors also found that targeting PDK4 m6A by dm6ACRISPR reduced PDK4 expression of PDK4 and consequently glycolysis in cancer cells.

Are they novel and will they be of interest to others in the field?

The authors tried to claim that their findings "for the first time" revealed "a crosstalk of mRNA m6A modification and cell metabolism", which is a rather overstatement. However, the findings do directly link m6A regulation to a glycolytic enzyme, which is interesting and innovative to certain extent

Is the work convincing, and if not, what further evidence would be required to strengthen the conclusions?

Overall, the studies are well designed, conducted and controlled. The data are in general supportive to the conclusions. However, there are some results that are somewhat preliminary and not terribly convincing. This reviewer has several major concerns:

- Most of the studies were solely relying on HeLa cells and this would not be appropriate. The

authors need to include additional human cervical cancer cell lines to repeat some key experiments in particular the “rescue” experiments of PDK4 in M3 knockdown cells.

- In addition, the authors need to include cell lines from other cancer types to address whether the effects of m6A on PDK4 expression and consequently glycolysis is a common phenomenon in diverse cancer types.
- What are the “erasers” (demethylases) for m6A regulation of CDK4? And how oncogenes or tumorigenic signals regulate/coordinate the writers/erasers/readers to regulate PDK4 expression? Is this true in highly proliferative cells, and when cells are stimulated for proliferation, what proliferative signals regulate m6A machinery to control PDK4?
- The clinical impact of the findings is not clear. The clinical study data presented did not support a correlation between the m6A status of PDK4 in cancer cells and disease burden/overall survival. The authors only checked for PDK4 protein expression levels, thus it is not convincing that PDK4 is regulated in cancers by increased m6A levels of PDK1 mRNA.

Specific comments:

- The data in Figure 1-2 are interesting but not very convincing that the metabolic changes of HeLa cells upon knockdown of M3 are primarily due to reduced PDK4 expression. Mettl3 deficiency probably affects a lot of glycolytic enzymes and mitochondrial proteins, and the most important “rescue” experiments shown in Fig 2J is not convincing because the overexpression levels of PDK4 in the M3-knockdown cells are way higher than the endogenous PDK4 (Fig. S2E). The authors need to include multiple cancer cell lines, not primarily relying on HeLa cells, to examine whether the “rescue” expression of PDK4 to a comparable level of endogenous PDK4 expression level would completely or at least predominantly reverse the metabolic phenotypes in cells with M3 knockdown.
- The results obtained using cells treated with dm6ACRISPR in Figure 5 need further analysis to determine whether removal of the m6A in PDK4 mRNA would result in decreased binding affinities of the “readers”, and whether the cells would be resistant to knockdown of M3 and the “readers” in terms of decreases in PDK4 mRNA and protein levels as well as consequently altered metabolic phenotypes.

On a more subjective note, do you feel that the paper will influence thinking in the field?

This reviewer feels that the reported findings lack in depth mechanistic insights regarding how oncogenes or tumorigenic signals regulate/coordinate the writers/erasers/readers to regulate PDK4 expression, and whether this occurs in normal proliferating cells that is “hijacked” by cancer cells.

Comment on the appropriateness of any statistical tests, and the accuracy of the description of any error bars and probability values

No major concerns

Disposition of Reviewers' Comments:

“N⁶-methyladenosine regulates metabolic reprogramming of cancer cells through PDK4”

(ID: NCOMMS-19-05918A-Z)

Addressing the points of Reviewer #1:

General: In this study, authors show the strong correlation between RNA m⁶A modification and cell metabolism. Both Mettl3 knockdown and ALHBH5 overexpression resulted in the decreased the glucose metabolism, lactate production and ATP levels. The only one target, PDK4, was identified from Mettl3 deleted Hela cells. The m⁶A sites in the 5'-UTR regulated PDK4 mRNA stability and translation by binding with IGF2BP3 and YTHDF1/eEF2 complex, respectively. Further, The m⁶A in the 5'-UTR was demethylated with dm6ACRISPR developed by their lab. This experiment confirmed the 6mA regulated PDK4 expression affected the glycolysis of cancer cells. **This work is novel and important for understanding the functional mechanism of mRNA m⁶A modification in cancer metabolism. For current version, the manuscript is well organized.** However, the authors need to provide more details and stronger evidence to support their conclusion.

Response: We sincerely thank the reviewer for the helpful comments and suggestions. Our itemised answers to questions and responses to reviewers as well as descriptions of how the manuscript was changed to address the comments are listed as below.

Specific comments

1 In Figure 2C, author claimed that the only candidate was identified from m⁶A-seq and mRNA-seq data. It was not clear that how the data was analyzed, which software, what is the cut-off?

Response: We thank the reviewer for the comment. As to m⁶A-seq, m⁶A enriched genes in wild type HeLa cells was analyzed in our previous study ¹ with the method described in supplementary data “m⁶A-seq data was analyzed according to protocols described before ².”

Significant peaks with FDR < 0.05 were annotated to RefSeq database (hg19). Sequence motifs were identified by using Homer. Gene expression was calculated by Cufflinks using sequencing reads from input samples. Cuffdiff was used to find DE genes.”

As to mRNA seq, the “Sequencing reads were mapped to reference human genome sequence (NCBI 36.1 [hg19] assembly by TopHat (Version 2.0.6). The differential expression between conditions was statistically assessed by R/Bioconductor package edgeR (version 3.0.8). Genes with FDR of < 0.05 and > 200 bp were called as differentially expressed.”

Among the 83 glucose metabolism related genes (Table S1), we identified the only candidate, pyruvate dehydrogenase kinase 4 (PDK4, Table S2), that overlapping among mRNA-seq (greater than 2.0-fold variation (p < 0.05) between wild type and *Mettl3^{Mut/-}* HeLa cells, 949 genes, Table S3) and m⁶A-seq (modification is more than 3 times greater than that in the input, p < 0.05, 4406 genes, Table S4) (Figure 2 C). We have provided detailed information in the revised manuscript

2 In Figure3, PDK4 mRNA stability and translation was decreased in *mettl3* deleted cells. Did authors observe this result in ALKBH5 overexpression cells?

Response: As suggested, we measured the mRNA stability and translation efficiency of PDK4 in ALKBH5 overexpression cells. Our data showed that over expression of ALKBH5 can also decrease the mRNA stability of PDK4 and translation efficiency of PDK4 reporter. This result further confirmed that m6A modification regulates the PDK4 mRNA stability and translation. The data have been added as Figure S3 C and D.

Figure R1 Effects of ALKBH5 on mRNA stability and translation efficiency of PDK4.

(A) HeLa cells were pre-transfected with vector control or ALKBH5 constructs for 24 h and

then treated with Act-D for the indicated times, the mature mRNA levels of PDK4 were checked;

- (B) HeLa cells were transfected with vector control or ALKBH5 constructs combined with pmirGLO-PDK4 reporter for 24 h. The translation outcome was determined as a relative signal of F-luc divided by R-luc, the mRNA abundance was determined by qRT-PCR of F-luc and R-luc, and the translation efficiency of PDK4 is defined as the quotient of reporter protein production (F-luc/R-luc) divided by mRNA abundance ².

3 How many m⁶A sites in the 5'-UTR of PDK4? In the targeted demethylation assay, did authors check the binding of YTHDF1 and IGF2BP3?

Response: m⁶A-RIP-seq data showed that one m⁶A peaks in 5'UTR region and three peaks in 3'UTR region (Figure 2D and Figure 4 A). To characterize m⁶A methylation in PDK4 mRNA, we used m⁶A antibody to immunoprecipitated fragmented RNA isolated from HeLa cells ¹. m⁶A-RIP-PCR showed that the m⁶A enrichment of PDK4 5'UTR was significantly down-regulated in *Mettl3*^{Mut/-} cells (Figure 4 B), suggesting that m⁶A methylation in 5'UTR regions might be more dynamic than those in 3'UTR.

According to the suggestion, we checked the binding of PDK4 mRNA with YTHDF1 and IGF2BP3 in cells treated with targeted demethylation assay. Our data showed that dm⁶ACRISPR with gRNA for PDK4 can significantly decrease the binding of PDK4 mRNA with YTHDF1 and IGF2BP3 (Figure R2). The data have been added as Figure 6 F.

Figure R2 m⁶A-RIP-qPCR analysis of PDK4 mRNA in HeLa cells transfected with dCas13b-ALKBH5 combined with gRNA control (dC-A5) or gRNA for PDK4 (dC-A5+gRNA) for 24 h by use of antibodies against YTHDF1 and IGF2BP3, respectively.

4 *In vivo* assays, I think the glycolysis parameters should be tested.

Response: To investigate whether PDK4 was involved in m⁶A regulated *in vivo* growth, HeLa wide type, *Mettl3*^{Mut/-}, HeLa^{PDK4}, and *Mettl3*^{Mut/- PDK4} stable cells were used to establish xenografts. Consistently, xenograft model confirmed that over expression of PDK4 can attenuate the suppression effect of *Mettl3*^{Mut/-} HeLa cells on *in vivo* growth (Figure 8 D). IHC showed that *Mettl3* depletion led to a lower level of PDK4 in xenograft tumor tissues (Figure 8 E), which suggested that the knockdown of *Mettl3* can regulate the expression of PDK4 *in vivo*.

We try to measure the glycolysis parameters such as the expression of IDH and GLUT among different groups by IHC, the results showed that the expression of IDH and GLUT among different groups had no significant variation since *Mettl3* had no effect on the expression of IDH or GLUT (data not shown).

According to the recent studies about glycolysis of cancer cells^{3, 4, 5, 6, 7, 8}, the major parameters are the tumor growth/volume and the expression of target protein. Overall, our data support the conclusion that PDK4 was involved in m⁶A regulated *in vivo* growth.

5 Most experiments were performed in the HeLa and Huh7 cell lines. They are derived from cervical cancer cells and liver cancer cells. The relationship of *Mettl3* expression level and disease survival is not consistent in different cancers. It is better to narrow the concept scope. Please refer to <https://www.proteinatlas.org/ENSG00000165819-Mettl3/pathology>.

Response: In our previous study, we found that deletion of *Mettl3* down-regulates m⁶A, impairs the migration, invasion and EMT of HeLa (cervical cancer), HepG2 and Huh7 (liver cancer) cells both *in vitro* and *in vivo*¹. Both sh-control and sh-*Mettl3* Huh7 cells were injected subcutaneously into nude female mice. Mice were sacrificed when the tumor volumes were about 100 mm³ for each group. IHC results showed that *Mettl3* depletion led to a lower level of Snail, FN, and vim in xenograft tumor tissues (Figure R3 A), which

suggested that the knockdown of *Mettl3* can decrease the metastasis potential of Huh7 cells when the tumor volumes were comparable. To determine the impacts of m⁶A methylation on *in vivo* metastasis, both sh-control and sh-*Mettl3* Huh7 cells were injected into the BALB/c nude mice, respectively, by tail vein injection to analyze lung colonization. Eight weeks after injection, experiment was terminated and lungs were analyzed to report the presence of metastatic tumors. As shown in Figure R3 B, the number of lung tumors derived from *Mettl3* knockdown Huh7 cells was significantly decreased compared to control cells, suggesting that *Mettl3* deficiency suppressed tumor metastasis *in vivo*. Further, the upregulation of *Mettl3* and YTHDF1 act as adverse prognosis factors for overall survival (OS) rate of liver cancer patients¹. All these data confirmed that m⁶A may play a positive role to promote the malignancy of liver cancer.

Figure R3 m⁶A methylation regulates *in vivo* cancer progression¹

In the present study, we also focused on roles of *Mettl3*/*PKD4* on progression of cervical and liver cancer. Over expression of *PKD4* can reverse the suppressed growth rate of *Mettl3*^{Mut/-} HeLa cells as compared to that of wild type cells (Figure 8 A). Consistently, *PKD4* also attenuated sh-*Mettl3* suppressed proliferation of Huh7 cells (Figure 8 B). Further, our data suggested that deletion of *Mettl3* increased sensitivity of HeLa cells to the treatment of doxorubicin (Dox), however, over expression of *PKD4* can attenuate this effect and reduce the Dox sensitivity (Figure 8 C). Xenograft model confirmed that over expression of *PKD4* can attenuate the suppression effect of *Mettl3*^{Mut/-} HeLa cells on *in vivo* growth (Figure 8 D).

Further, **according to the suggestions of reviewer 2, we knocked down the expression of *Mettl3* in SiHa cell, another cervical cancer cell line.** The key experiments including

Mettl3 on the expression and mRNA half-lives of PDK4, “rescue” experiments of PDK4 in Mettl3 knockdown cells about metabolic reprogramming and cell proliferation, have been conducted. Our data confirmed that knocked down the expression of Mettl3 can suppress the expression of PDK4 in SiHa cells (Figure S2 D). Further, over expression of PDK4 (Figure S2 F) can reverse deletion of Mettl3 suppressed glucose consumption, lactate production, and ATP generation of SiHa cells (Figure S2 G). Consistently, deletion of Mettl3 can decrease the half-lives of PDK4 mRNA in SiHa cells (Figure S3 B). Over expression of PDK4 (Figure S8 C) can also attenuate si-Mettl3 suppressed proliferation of SiHa cells (Figure S8 D) . **All these results have been added in the revised manuscript.**

Recent studies indicated that Mettl3 plays an oncogenic role for liver cancer ⁹. Mettl3 depletion results in strong inhibition of cancer cell growth including HeLa cells ¹⁰. It also exhibits several functions in cancer cells, such as: maintaining myeloid leukemia ¹¹, playing essential role in GSC maintenance and radioresistance ¹², positively regulating proliferation of hematopoietic stem/progenitor cells (HSPCs) ¹³ and triggering the translation in cancer cells ¹⁰. Contradictorily, Mettl3 acts as a tumor suppressor in renal cell carcinoma ¹⁴.

According to the suggestions, we have narrowed the concept scope to cervical and liver cancers in all parts of the revised manuscript including abstract, introduction, results and discussion.

6 In Figure 7J, What does the curve means ? Is it Mettl3 with low PDK4 or High PDK4? Whether the two genes have synergic effect on the disease free survival?

Response: Using the online bioinformatics tools including GEPIA and Kaplan-Meier plotter, we found that cervical cancer patients with increased expression of Mettl3 showed reduced disease-free survival (DFS, Figure 8 J). **We have corrected the mislabeling of the previous figure.**

According to the suggestion, we further analyzed whether these two genes have synergic effect on the clinical prognosis of cervical cancer patients. As shown in Figure R4, cervical cancer patients with increased expression of PDK4 and Mettl3 showed significant reduced overall survival (OS). Since the sample number is less than 50, the data have not

been added in the revised manuscript.

Figure R4 The Kaplan-Meier survival curves of OS based on Mettl3/PDK4 expression in cervical cancer patients from TCGA data base.

Addressing the points of Reviewer #2:

General: The authors reported findings using HeLa cells that mitochondrial PDK4 is positively regulated by m6A modified 5'-UTR, leading to enhanced translation elongation and mRNA stability through binding with m6A “readers” including YTHDF1/eEF-2 complex and IGF2BP3. The authors also found that targeting PDK4 m6A by dm6ACRISPR reduced PDK4 expression of PDK4 and consequently glycolysis in cancer cells.

Response: Thank you so much for the review. The suggestions are very helpful for the revision of manuscript. Our itemised answers to questions and responses to reviewers as well as descriptions of how the manuscript was changed to address the comments are listed as below.

Are they novel and will they be of interest to others in the field?

The authors tried to claim that their findings “for the first time” revealed “a crosstalk of mRNA m6A modification and cell metabolism”, which is a rather overstatement. However, **the findings do directly link m6A regulation to a glycolytic enzyme, which is interesting and innovative to certain extent.**

Response: According to the suggestion, we have revised the title to “*N*⁶-methyladenosine

regulates metabolic reprogramming of cancer cells through PDK4". In addition, the conclusion has been revised to "Our study reveals that m⁶A regulates metabolic reprogramming of cancer cells through induction of PDK4, which broadens the multilayer regulation of metabolism and expands our understanding of such interplays that are essential for therapeutic application". We have checked the whole manuscript including introduction, results and discussion to revise the overstatement. Specifically, all "for the first time" and "a crosstalk of mRNA m6A modification and cell metabolism" have been revised.

Is the work convincing, and if not, what further evidence would be required to strengthen the conclusions?

Overall, the studies are well designed, conducted and controlled. The data are in general supportive to the conclusions. However, there are some results that are somewhat preliminary and not terribly convincing. This reviewer has several major concerns:

- Most of the studies were solely relying on HeLa cells and this would not be appropriate. The authors need to include additional human cervical cancer cell lines to repeat some key experiments in particular the "rescue" experiments of PDK4 in M3 knockdown cells.

Response: According to the suggestion, we knocked down the expression of Mettl3 in SiHa cell, another cervical cancer cell line. The key experiments including Mettl3 on the expression and mRNA half-lives of PDK4, "rescue" experiments of PDK4 in Mettl3 knockdown cells about metabolic reprogramming and cell proliferation, have been conducted.

Our data confirmed that knocked down the expression of Mettl3 can suppress the expression of PDK4 in SiHa cells (Figure S2 D). Further, over expression of PDK4 (Figure S2 F) can reverse deletion of Mettl3 suppressed glucose consumption, lactate production, and ATP generation of SiHa cells (Figure S2 G). Consistently, deletion of Mettl3 can decrease the half-lives of PDK4 mRNA in SiHa cells (Figure S3 B). Over expression of PDK4 (Figure S8 C) can also attenuate si-Mettl3 suppressed proliferation of SiHa cells (Figure S8 D). All these results have been added in the revised manuscript.

- In addition, the authors need to include cell lines from other cancer types to address whether the effects of m⁶A on PDK4 expression and consequently glycolysis is a common phenomenon in diverse cancer types.

Response: As mentioned in the response to the Reviewer #1, Mettl3 may play both oncogenic and anti-oncogenic roles. Recent studies indicated that Mettl3 plays an oncogenic role for liver cancer ⁹. Mettl3 depletion results in strong inhibition of cancer cell growth including HeLa cells ¹⁰. It also exhibits several functions in cancer cells, such as: maintaining myeloid leukemia ¹¹, playing essential role in GSC maintenance and radioresistance ¹², positively regulating proliferation of hematopoietic stem/progenitor cells (HSPCs) ¹³ and triggering the translation in cancer cells ¹⁰. Contradictorily, Mettl3 acts as a tumor suppressor in renal cell carcinoma ¹⁴.

In our previous study, we found that deletion of Mettl3 down-regulates m⁶A, impairs the migration, invasion and EMT of **HeLa (cervical cancer) , HepG2 and Huh7 (liver cancer)** cells both *in vitro* ¹. In the present study, we also focused on roles of Mettl3/PKD4 on progression of cervical and liver cancer. Although our data showed that knockdown of *Mettl3* can inhibit the ATP generation of **MDA-MB-231 cells** (Figure S1 C), **according to the suggestions, we have narrowed the concept scope to cervical and liver cancers in all parts of the revised manuscript including abstract, introduction, results and discussion on the basis of data from HeLa, SiHa, Huh7, and HepG2 cells.**

- What are the “erasers” (demethylases) for m⁶A regulation of PDK4?

Response: m⁶A demethylases FTO (also known as ALKBH9) ¹⁵ and ALKBH5 ¹⁶ are responsible for demethylation of mRNA m⁶A in cell nucleus. Recently, the role of FTO as a specific eraser of mRNA m⁶A has recently been challenged. In 2017, Mauer et al. ¹⁷ reported that the substrate of FTO was likely to be m⁶Am with the demethylation activity of FTO was approximately 100 times higher towards m⁶Am than towards m⁶A. Recently, Mauer et al. ¹⁸ reported that RNA demethylase FTO selectively demethylates the m⁶Am during the biogenesis of small nuclear RNAs (snRNAs). The distinct roles of FTO in RNA demethylation may be dependent on the subcellular localization of FTO and its substrate RNAs, the internal or cap position of methylated adenosine, and the species of RNA such

as mRNA, snRNA, and tRNA¹⁹. While ALKBH5 has been revealed as the mammalian RNA demethylase located in the nuclear speckles to remove the m⁶A modification both *in vitro* and *in vivo*¹⁶.

We found that over expression of ALKBH5 can suppress the expression of PDK4 in both HeLa and Huh7 cells (Figure 2 G). Over expression of m⁶A demethylase ALKBH5 in HeLa cells (Figure S1 D) can also decrease the glucose consumption, lactate production rate, and ATP levels of HeLa cells (Figure 1 H). Consistently, ALKBH5 can increase OCR while decrease ECAR in HeLa cells (Figure S1 H and I). In addition, over expression of ALKBH5 can decrease the mature mRNA stability of PDK4 in HeLa cells (Figure S3 C). The translation efficiency of PDK4 was also reduced in cells co-transfected with ALKBH5 constructs (Figure S3 D). In the part of targeting m⁶A of PDK4 by dm⁶ACRISPR, our data showed that gRNAs combined with dCas13b-ALKBH5 can significantly decrease the m⁶A levels of targeted site of PDK4 mRNA (Figure 6 C). The dCas13b-ALKBH5 mediated demethylation of PDK4 was further confirmed by m⁶A-RIP-qPCR (Figure S6 D). Together, **all these data suggested that ALKBH5 should be the demethylase of PDK4.**

We have tested the effect of FTO on the expression of PDK4. As showed in Figure R5, over expression of FTO had no effect on the protein and mRNA expression of PDK4 in HeLa cells. In addition, it also had no effect on the mRNA stability of PDK4 in HeLa cells (Figure R5 C). **These results suggested that FTO might not the demethylase of PDK4 in HeLa cells.**

Figure R5 Over expression of FTO on the protein (A), mRNA (B) or mRNA stability of PDK4 in HeLa cells.

And how oncogenes or tumorigenic signals regulate/coordinate the writers/erasers/readers to regulate PDK4 expression? Is this true in highly proliferative cells, and when cells are stimulated for proliferation, what proliferative signals regulate m⁶A machinery to control PDK4?

Response: We evaluated the potential mechanisms responsible for the expression of Mettl3 in cancer cells. The expression of Mettl3 in ECT1/E6E7, a human normal cervical epithelium cell line, and cervical cancer HeLa and SiHa cells were checked. Our data showed that the protein (Figure 7 A) and mRNA (Figure 7 B) expression of Mettl3 was increased in cervical cancer HeLa and SiHa cells as compared with that in ECT1/E6E7 cells. In addition, the precursor mRNA of Mettl3 in cervical cancer cells was also greater than that in ECT1/E6E7 cells (Figure 7 C). Further, we generated the promoter reporter of Mettl3 via insertion the upstream 1 Kb upstream of the transcription start site (TSS) of Mettl3 to pGL3 plasmid. The luciferase assay showed that the promoter activities of Mettl3 in cervical cancer cells were significantly greater than that in ECT1/E6E7 cells (Figure 7 D). It indicated that the transcription of Mettl3 was activated in cervical cancer cells.

We then evaluated the potential transcription factors (TFs) responsible for the regulation of Mettl3 in cancer cells. To identify TFs that directly regulate Mettl3 expression, we analyzed the ENCODE chromatin immunoprecipitation sequencing (ChIP-seq) data in ChIPBase²⁰ and PROMO with 5% maximum matrix dissimilarity rate²¹. Among the 47 factors identified by ChIPBase and 32 factors identified by PROMO, eight factors including ETS1, FOXA1, NRF1, PAX5, STAT4, TBP, TP53, and YY1 were overlapping between the two databases (Figure 7 E). We further compared the expression of these factors among ECT1/E6E7 and cervical cancer cells. Our data showed that only mRNA of NRF1 and TBP were upregulated in both HeLa and SiHa cells as compared with that in ECT1/E6E7 cells (Figure 7F). Western blot analysis showed that the expression of TBP, rather than NRF1, in cervical cancer cells was significantly greater than that in ECT1/E6E7 cells (Figure 7 G). We then knocked down the expression of TBP in HeLa and SiHa cells by use of its specific siRNA (Figure S7 A). Our data showed that

knockdown of TBP can suppress the mRNA (Figure 7 H) and protein (Figure 7 I and Figure S7 A) expression of Mettl3 in both HeLa and SiHa cells.

Chromatin immunoprecipitation (ChIP)-qPCR assays demonstrated that TBP has a significant enrichment of Mettl3 promoter over normal immunoglobulin G (IgG) control in both HeLa and SiHa cells (Figure 7 J), indicating a direct binding between TBP and Mettl3 promoter. ChIPBase data (Figure S7 B)²² and JASPAR (Figure S7 C) showed that there were two TBP potential binding sites in the promoter within 1 Kb upstream of Mettl3. ChIP-qPCR showed that binding of TBP to the potential binding site 2 was much greater than to the site 1 in HeLa cells (Figure 7 K). We then mutant the two TBP potential binding sites of promoter reporter of Mettl3 to generate the pGL-M3-Mut1 or pGL-M3-Mut2 (Figure S7 D and Figure 7 L). Our data showed that si-TBP can significantly decrease luciferase of pGL-M3-WT and pGL-M3-Mut1, while the inhibition effect of si-TBP was attenuated for pGL-M3-Mut2 (Figure 7 M). Consistently, the relative values of F-Luc/R-luc of pGL-M3-WT and pGL-M3-Mut1 in HeLa cells were greater than that in ETC1/E6E7 cells, while this effect was attenuated for pGL-M3-Mut2 (Figure S7 E). Further, the expression of TBP was significantly positively correlated with the expression of Mettl3 in clinical cervical cancer patients from ChIPBase²⁰ (Figure 7 N), GEPIA (Figure S7 F), or TCGA database (Figure S7 G). **All these data suggested that TBP might be responsible for the upregulation of Mettl3 in cervical cancer cells via binding to its promoter-proximal site to increase its transcription.**

Figure 7 TBP is responsible for the upregulation of Mettl3 in cervical cancer cells

(A~C) The protein (A), mature mRNA (B), or precursor mRNA (C) of Mettl3 in ECT1/E6E7, HeLa, and SiHa cells were checked;

(D) The promoter activity of Mettl3 in ECT1/E6E7, HeLa, and SiHa cells were checked by dual-luciferase assay;

(E) Venn diagram shows the overlap of transcription factors of Mettl3 predicted by PROMO and ChIPBase, respectively;

(F) The mRNA expression of potential transcription factors of Mettl3 in ECT1/E6E7, HeLa, and SiHa cells were checked by PCR analysis;

(G) The protein expression of NRF1 and TBP was checked by western blot analysis;

(H&I) Cells were transfected by siRNA negative control (si-NC) or siRNAs of TBP for 24 h,

the mRNA (H) and protein (I) of *Mettl3* were checked;

- (J) The binding between TBP and promoter of *Mettl3* was checked by ChIP-PCR using IgG or TBP antibody;
- (K) Binding between TBP transcriptional factor and the promoter of *Mettl3* at the potential binding site “1” and “2” or negative site “BLK” was checked by ChIP-PCR;
- (L) Schematic representation of the mutated promoter in pGL3-Basic-*Mettl3*-luc reporter to investigate the role of TBP in *Mettl3* expression;
- (M) HeLa cells were co-transfected with pGL3-*Mettl3*-WT-Luc, pGL3-*Mettl3*-Mut1-Luc, pGL3-*Mettl3*-Mut2-Luc, pRL-TK plasmid and si-NC or si-TBP-1 for 24 h. Results were expressed as the ratio between the activity of the reporter plasmid and pRL-TK;
- (N) Correlation between *Mettl3* and *TBP* in cervical cancer patients (n=309) from ChIPBase database.

Data are presented as the mean \pm SD from three independent experiments. * $p < 0.05$, ** $p < 0.01$.

NS, no significant.

Figure S7 TBP is responsible for the upregulation of Mettl3 in cervical cancer cells

- (A) SiHa cells were transfected by siRNA negative control (si-NC) or siRNAs of TBP for 24 h, the protein of Mettl3 were checked;
- (B) The binding motif of TBP analyzed by ChIPBase data;
- (C) The potential binding sites of TBP in the promoter of Mettl3 were analyzed by use of JASPAR;
- (D) Schematic representation of mutation in promoter to investigate the effects of TBP on transcription of Mettl3;
- (E) Cells were co-transfected with pGL3-Mettl3-WT-Luc, pGL3-Mettl3-Mut1-Luc,

pGL3-Mettl3-Mut2-Luc, and pRL-TK plasmid for 24 h. Results were expressed as the ratio between the activity of the reporter plasmid and pRL-TK;

(F&G) Correlation between *Mettl3* and *TBP* in cervical cancer patients from GEPIA (n=146) and TCGA (n=169) database.

Data are presented as the mean \pm SD from three independent experiments. *p<0.05, NS, no significant.

Related to Figure 7

Further, we have added discussion about this part “Finally, we found that TBP can regulate the transcription of *Mettl3* to increase its expression in cervical cancer cells, which was evidenced by the results of ChIP-PCR analysis and luciferase reporter assay. Nowadays, transcription factors which can regulate the expression of RNA methyltransferases are not well illustrated. It has been reported that the expression of *Mettl14* is negatively regulated by SPI1 in acute myeloid leukemia (AML) cells²³. As to *Mettl3*, hepatitis B X-interacting protein (HBXIP) can increase its expression in breast cancer cells via inhibiting miRNA let-7g, which down-regulated the expression of *Mettl3* by targeting its 3'UTR²⁴. MiR-33a can decrease the expression of *Mettl3* in lung cancer cells to suppress cell proliferation²⁵. In gastric cancer cells, H3K27ac can activate *Mettl3* transcription to increase its expression²⁶. Our data provided the novel mechanisms responsible for the upregulation of *Mettl3* in cancer cells.”

We further investigated the potential effects of m⁶A on the PDK4 expression and metabolic programming of ECT1/E6E7, a human normal cervical epithelium cell line. The data showed that in normal cervical epithelium cells, the effect of *Mettl3* deletion induced suppression of PDK4 and inhibition of glucose consumption and ATP generation were markedly attenuated as compared with that effects in HeLa cells (Figure R6 A to C). This might be due to that PDK4 mRNA was less methylated by m⁶A in ECT1/E6E7 cells due to the lower levels of *Mettl3* (Figure 7A). Therefore deletion of *Mettl3* had less effect on the expression of PDK4 and metabolic characteristics.

Figure R6 The mRNA expression (A), glucose consumption (B), and ATP levels (C) of HaLa and ECT1 cells with or without knockdown of Mettl3.

The clinical impact of the findings is not clear. The clinical study data presented did not support a correlation between the m6A status of PDK4 in cancer cells and disease burden/overall survival. The authors only checked for PDK4 protein expression levels, thus it is not convincing that PDK4 is regulated in cancers by increased m6A levels of PDK4 mRNA.

Response: In our xenograft study, HeLa wide type, *Mettl3*^{Mut/-}, HeLa^{PDK4}, and *Mettl3*^{Mut/- PDK4} stable cells were used to establish xenografts. Consistently, xenograft model confirmed that over expression of PDK4 can attenuate the suppression effect of *Mettl3*^{Mut/-} HeLa cells on *in vivo* tumor growth (Figure 8 D). IHC showed that Mettl3 depletion led to a lower level of PDK4 in xenograft tumor tissues (Figure 8 E), which suggested that the knockdown of Mettl3 can regulate the expression of PDK4 *in vivo*. **It revealed that PDK4 mediates Mettl3-regulated *in vivo* growth of cancer cells.**

According to the suggestions, we questioned the possibility of a link between m⁶A methylation, PDK4 and cancer development. **The expression of Mettl3 was positively correlated with the PDK4 mRNA in cervical cancer patients (Figure 8 F).** Further, Mettl3 expression in cervical cancer tissues was significantly (p<0.01) greater than that in normal tissues, according to Zhai Cervix, Biewenga Cervix, and Pyeon Muti-Cancer data from the Oncomine database (Figure 8 G). **Consistently, IGF2BP3 (Figure 8 H) and YTHDF1 (Figure 8 I), the readers of PDK4 mRNA, were significantly increased in cervical cancer tissues as compared with that in the control samples.**

Using the online bioinformatics tools including GEPIA and Kaplan-Meier plotter, we found that cervical cancer patients with increased expression of **Mettl3 (Figure 8 J)** and **TBP (Figure 8 K)** showed reduced disease-free survival (DFS). Consistently, cervical cancer patients with increased expression of PDK4 showed reduced disease-free survival (DFS, Figure 8 L) and overall survival (OS, Figure 8 M).

Since we have proved that 1) PDK4 mediates Mettl3-regulated both *in vitro* and *in vivo* growth of cancer cells; 2) m⁶A related proteins (Mettl3, IGF2BP3, and YTHDF1) and PDK4 were upregulated in cervical cancer tissues than that in normal tissues; 3) The mRNA of Mettl3 was correlated with the expression of PDK4 in cervical cancer tissues; and 4) m⁶A related proteins (Mettl3, IGF2BP3, and YTHDF1) and PDK4 are correlated with poor prognosis of cervical cancer patients. We think all these data can support our conclusion that PDK4 is involved in m⁶A regulated metabolic reprogramming and cancer progression.

Specific comments

The data in Figure 1-2 are interesting but not very convincing that the metabolic changes of HeLa cells upon knockdown of M3 are primarily due to reduced PDK4 expression. Mettl3 deficiency probably affects a lot of glycolytic enzymes and mitochondrial proteins, and the most important “rescue” experiments shown in Fig 2J is not convincing because the overexpression levels of PDK4 in the M3-knockdown cells are way higher than the endogenous PDK4 (Fig. S2E). The authors need to include multiple cancer cell lines, not primarily relying on HeLa cells, to examine whether the “rescue” expression of PDK4 to a comparable level of endogenous PDK4 expression level would completely or at least predominantly reverse the metabolic phenotypes in cells with M3 knockdown.

Response: According to the suggestions, we decreased the transfection amount of PDK4 constructs to a comparable level of endogenous PDK4 expression level in the “rescue” experiments. Our data showed that the less over expression of PDK4 (Figure S2 F) can reverse the down regulation of glucose consumption, lactate production rate, and ATP levels (Figure 2 J). In addition, over expression of PDK4 constructs to a comparable level of endogenous PDK4 expression level can also attenuate the down regulation of

glucose consumption, lactate production rate, and ATP levels in Mettl3 knocked down SiHa cells (**Figure S2G**). All these data confirmed that PDK4 was involved in Mettl3 regulated metabolic reprogramming of cervical cancer cells.

In addition, over expression of PDK4 can reverse the suppressed growth rate of Mettl3 deleted HeLa (Figure 8 A) and SiHa (Figure S8 D) cells as compared to that of their corresponding control cells. Consistently, PDK4 also attenuated sh-Mettl3 suppressed proliferation of Huh7 cells (Figure 8 B). To investigate whether PDK4 was involved in m⁶A regulated *in vivo* growth, HeLa wide type, *Mettl3*^{Mut/-}, HeLa^{PDK4}, and *Mettl3*^{Mut/- PDK4} stable cells were used to establish xenografts. Consistently, xenograft model confirmed that over expression of PDK4 can attenuate the suppression effect of *Mettl3*^{Mut/-} HeLa cells on *in vivo* growth (Figure 8 D). All these data suggested that PDK4 mediates Mettl3-regulated metabolic reprogramming and growth of cervical cancer cells

The results obtained using cells treated with dm6ACRISPR in Figure 5 need further analysis to determine whether removal of the m6A in PDK4 mRNA would result in decreased binding affinities of the “readers”, and whether the cells would be resistant to knockdown of M3 and the “readers” in terms of decreases in PDK4 mRNA and protein levels as well as consequently altered metabolic phenotypes.

Response: According to the suggestions, we analyzed the binding between IGF2BP3 and YTHDF1 with PDK4 mRNA in cells treated with dm6ACRISPR or control conditions. Our data confirmed that dm6ACRISPR can significantly decrease the binding between IGF2BP3 and PDK4 mRNA and between Mettl3 with PDK4 mRNA (Figure 6 F). Further, in dm6ACRISPR treated HeLa cells, the further deletion of Mettl3 had limited effect on the expression of PDK4 (Figure R7 A) and metabolic phenotypes including glucose consumption (Figure R7 B).

Figure R7. HeLa cells transfected with dCas13b-ALKBH5 combined with gRNA control or gRNA for PDK4 combined with si-NC or si-Mettl3 for 24 h. The mRNA of PDK4 (A) and glucose consumption (B) were measured, respectively.

On a more subjective note, do you feel that the paper will influence thinking in the field?

This reviewer feels that the reported findings lack in depth mechanistic insights regarding how oncogenes or tumorigenic signals regulate/coordinate the writers/erasers/readers to regulate PDK4 expression, and whether this occurs in normal proliferating cells that is “hijacked” by cancer cells.

Response: According to the suggestions, we have further investigated mechanisms responsible for the upregulation of Mettl3 in cervical cancer cells. **Our data showed that the upregulation of TBP in cervical cancer cell can directly initiate the transcription of Mettl3 in cancer cells (Figure 7 and Figure S7),** which can further methylate PDK4 mRNA to increase its mRNA stability and translation.

We further investigate the potential effects of m6A on the PDK4 expression and metabolic programming of ECT1/E6E7, a human normal cervical epithelium cell line. The data showed that in normal cervical epithelium cells, the effect of Mettl3 deletion induced suppression of PDK4 and inhibition of glucose consumption and ATP generation were markedly attenuated as compared with that effects in HeLa cells (Figure R6 A to C). This might be due to that PDK4 mRNA was less methylated by m6A in

ECT1/E6E7 cells due to the lower levels of Mettl3 (Figure 7A). Therefore deletion of Mettl3 had less effect on the expression of PDK4 and metabolic characteristics.

The revised manuscript will influence thinking in the field including: 1) we identified for the first time that m⁶A can regulate the glycolysis and ATP generation of cancer cells via regulation of PDK4; 2) The m⁶A of PDK4 5'UTR regulates the mRNA stability and translation of PDK4 via binding with IGF2BP3 and YTHDF1, respectively; 3) For the first time, we specifically demethylated the m⁶A of PDK4 mRNA by use of dm⁶ACRISPR system and revealed it can decrease the expression of PDK4 and cellular glycolysis; 4) In cervical cancer cells, we identified that TBP is the transcription factor responsible for the upregulation of Mettl3, which can broaden the limited information about the transcriptional regulation of Mettl3 in cancer cells.

As to mechanistic link for m⁶A/Mettl3 regulated expression of PDK4:

- 1) On the basis of sequencing and functional study, we found that PDK4 is the most important m⁶A/Mettl3 regulated genes involved in cellular metabolic. m⁶A/Mettl3 can regulate the expression of PDK4 both *in vitro* and *in vivo* (Figure 2, 3 and 8);
- 2) We excluded the effect of m⁶A on transcription (promoter activity), nuclear turnover, cap-dependent translation, and protein stability of Snail. Our data revealed that m⁶A promotes the expression of PDK4 only via increasing its translation elongation and mRNA stability;
- 3) We confirmed that the m⁶A on "GGAC" located at 5'UTR while not in 3'UTR mediated the m⁶A triggered expression of PDK4;
- 4) We found that the m⁶A can increase the binding between YTHDF1 and IGF2BP3 to increase the translation and mRNA stability of PDK4 mRNA, respectively;
- 5) We specifically demethylate PDK4 mRNA to confirm its roles in m⁶A regulated metabolic reprogram. To our knowledge, it is the first time to demethylate specific mRNA to investigate its biological function.

Contrary to many other papers on this topic that do not even verify the specific methylation site, excluding the other processes of mRNA, and illustrating the detailed translation mechanisms on m⁶A regulated gene expression (Cell Stem Cell. 2018; 22(2):191-205;

Cell Res. 2018 Sep;28(9):955-957; Hepatology 2018; 67(6):2254-2270; Cancer Cell. 2017 10;31(4):591-606, also including our recent study Nat Commun. 2018 14;9(1):4772)^{23, 27, 28, 29, 30}, we have provided strong evidences about the Mettl3 mediated m⁶A modification in regulation of PKD4 expression. Reviewer #1 also highly appraised our detailed mechanisms for the m⁶A regulated gene expression/translation: “This work is novel and important for understanding the functional mechanism of mRNA m6A modification in cancer metabolism. For current version, the manuscript is well organized.”

Together, our study provided evidence that RNA methylation can regulate the metabolic reprogramming of cancer cells via regulating PDK4. We anticipate that this work provokes considerable efforts in the epitranscriptome field that will address how common this mode of regulation is in different cancer types.

Reference

1. Lin X, *et al.* RNA m(6)A methylation regulates the epithelial mesenchymal transition of cancer cells and translation of Snail. *Nat Commun* **10**, 2065 (2019).
2. Wang X, *et al.* N(6)-methyladenosine Modulates Messenger RNA Translation Efficiency. *Cell* **161**, 1388-1399 (2015).
3. Ling S, *et al.* USP22 promotes hypoxia-induced hepatocellular carcinoma stemness by a HIF1alpha/USP22 positive feedback loop upon TP53 inactivation. *Gut*, (2019).
4. Liang C, *et al.* Localisation of PGK1 determines metabolic phenotype to balance metastasis and proliferation in patients with SMAD4-negative pancreatic cancer. *Gut*, (2019).
5. Wong TL, *et al.* CRAF methylation by PRMT6 regulates aerobic glycolysis driven hepatocarcinogenesis via ERK-dependent PKM2 nuclear relocalization and activation. *Hepatology*, (2019).
6. Zhou L, *et al.* HOXA9 inhibits HIF-1alpha-mediated glycolysis through interacting with CRIP2 to repress cutaneous squamous cell carcinoma development. *Nat Commun* **9**, 1480 (2018).
7. Humpton TJ, *et al.* Oncogenic KRAS Induces NIX-Mediated Mitophagy to Promote Pancreatic Cancer. *Cancer Discov* **9**, 1268-1287 (2019).

8. Fang Y, *et al.* CD36 inhibits beta-catenin/c-myc-mediated glycolysis through ubiquitination of GPC4 to repress colorectal tumorigenesis. *Nat Commun* **10**, (2019).
9. Chen M, *et al.* RNA N6-methyladenosine methyltransferase METTL3 promotes liver cancer progression through YTHDF2 dependent post-transcriptional silencing of SOCS2. *Hepatology*, (2017).
10. Lin SB, Choe J, Du P, Triboulet R, Gregory RI. The m(6)A Methyltransferase METTL3 Promotes Translation in Human Cancer Cells. *Molecular Cell* **62**, 335-345 (2016).
11. Barbieri I, *et al.* Promoter-bound METTL3 maintains myeloid leukaemia by m(6)A-dependent translation control. *Nature* **552**, 126-131 (2017).
12. Visvanathan A, *et al.* Essential role of METTL3-mediated m(6)A modification in glioma stem-like cells maintenance and radioresistance. *Oncogene* **37**, 522-533 (2018).
13. Vu LP, *et al.* The N(6)-methyladenosine (m(6)A)-forming enzyme METTL3 controls myeloid differentiation of normal hematopoietic and leukemia cells. *Nat Med* **23**, 1369-1376 (2017).
14. Li X, *et al.* The M6A methyltransferase METTL3: acting as a tumor suppressor in renal cell carcinoma. *Oncotarget* **8**, 96103-96116 (2017).
15. Jia G, *et al.* N6-methyladenosine in nuclear RNA is a major substrate of the obesity-associated FTO. *Nature Chem Biol* **7**, 885-887 (2011).
16. Zheng G, *et al.* ALKBH5 is a mammalian RNA demethylase that impacts RNA metabolism and mouse fertility. *Mol Cell* **49**, 18-29 (2013).
17. Mauer J, *et al.* Reversible methylation of m(6)Am in the 5' cap controls mRNA stability. *Nature* **541**, 371-375 (2017).
18. Mauer J, *et al.* FTO controls reversible m(6)Am RNA methylation during snRNA biogenesis. *Nat Chem Biol* **15**, 340-347 (2019).
19. Wei J, *et al.* Differential m(6)A, m(6)Am, and m(1)A Demethylation Mediated by FTO in the Cell Nucleus and Cytoplasm. *Mol Cell* **71**, 973-985 e975 (2018).
20. Zhou KR, *et al.* CHIPBase v2.0: decoding transcriptional regulatory networks of non-coding RNAs and protein-coding genes from ChIP-seq data. *Nucleic Acids Res* **45**, D43-D50 (2017).
21. Messeguer X, Escudero R, Farre D, Nunez O, Martinez J, Alba MM. PROMO: detection of known transcription regulatory elements using species-tailored searches. *Bioinformatics* **18**, 333-334 (2002).

22. Heinz S, *et al.* Simple combinations of lineage-determining transcription factors prime cis-regulatory elements required for macrophage and B cell identities. *Mol Cell* **38**, 576-589 (2010).
23. Weng H, *et al.* METTL14 Inhibits Hematopoietic Stem/Progenitor Differentiation and Promotes Leukemogenesis via mRNA m(6)A Modification. *Cell Stem Cell* **22**, 191-205 e199 (2018).
24. Cai X, *et al.* HBXIP-elevated methyltransferase METTL3 promotes the progression of breast cancer via inhibiting tumor suppressor let-7g. *Cancer Lett* **415**, 11-19 (2018).
25. Du M, *et al.* MiR-33a suppresses proliferation of NSCLC cells via targeting METTL3 mRNA. *Biochem Biophys Res Commun* **482**, 582-589 (2017).
26. Wang Q, *et al.* METTL3-mediated m(6)A modification of HDGF mRNA promotes gastric cancer progression and has prognostic significance. *Gut*, (2019).
27. Jin S, *et al.* m(6)A RNA modification controls autophagy through upregulating ULK1 protein abundance. *Cell Res* **28**, 955-957 (2018).
28. Chen M, *et al.* RNA N6-methyladenosine methyltransferase-like 3 promotes liver cancer progression through YTHDF2-dependent posttranscriptional silencing of SOCS2. *Hepatology* **67**, 2254-2270 (2018).
29. Zhang S, *et al.* m6A Demethylase ALKBH5 Maintains Tumorigenicity of Glioblastoma Stem-like Cells by Sustaining FOXM1 Expression and Cell Proliferation Program. *Cancer Cell* **31**, 591-606.e596 (2017).
30. Wu Y, *et al.* Mettl3-mediated m(6)A RNA methylation regulates the fate of bone marrow mesenchymal stem cells and osteoporosis. *Nat Commun* **9**, 4772 (2018).

REVIEWERS' COMMENTS:

Reviewer #1 (Remarks to the Author):

The authors have added up more experimental results and addressed most of our questions in this version.

Only one minor issue here:

1. ECAR in the text, ECR in the figure 1D, 1F.

Please correct it.

Reviewer #2 (Remarks to the Author):

The authors performed substantial experiments and satisfactorily addressed all the concerns of this reviewer. The revised manuscript is very much strengthened to be a strong candidate for Nature Communications

REVIEWERS' COMMENTS:

Reviewer #1 (Remarks to the Author)

Comments: The authors have added up more experimental results and addressed most of our questions in this version.

Only one minor issue here:

1. ECAR in the text, ECR in the figure 1D, 1F.

Please correct it.

Response/Action: The authors appreciate the review and comments, which are very helpful to improve the manuscript. Figure 1 D and 1 F have been revised to “ECAR”.

Reviewer #2 (Remarks to the Author)

Comments: The authors performed substantial experiments and satisfactorily addressed all the concerns of this reviewer. The revised manuscript is very much strengthened to be a strong candidate for Nature Communications.

Response/Action: The authors appreciate the review and comments, which are very helpful to improve the manuscript.